# Cheaply Estimating Inference Efficiency Metrics for Autoregressive Transformer Models

**Deepak Narayanan**[*]
NVIDIA
dnarayanan@nvidia.com

**Keshav Santhanam**
Stanford University
keshav2@cs.stanford.edu

**Peter Henderson**
Stanford University
phend@cs.stanford.edu

**Rishi Bommasani**
Stanford University
nlprishi@stanford.edu

**Tony Lee**
Stanford University
tonyhlee@stanford.edu

**Percy Liang**
Stanford University
pliang@cs.stanford.edu

## Abstract

Large language models (LLMs) are highly capable but also computationally expensive. Characterizing the *fundamental tradeoff* between inference efficiency and model capabilities is thus important, but requires an efficiency metric that is comparable across models from different providers. Unfortunately, raw runtimes measured through black-box APIs do not satisfy this property: model providers can implement software and hardware optimizations orthogonal to the model, and shared infrastructure introduces performance contention. We propose a new metric for inference efficiency called *idealized runtime*, that puts models on equal footing as though they were served on uniform hardware and software without performance contention, and a cost model to efficiently estimate this metric for autoregressive Transformer models. We also propose variants of the idealized runtime that incorporate the number and type of accelerators needed to serve the model. Using these metrics, we compare ten LLMs developed in 2022 to provide the first analysis of inference efficiency-capability tradeoffs; we make several observations from this analysis, including the fact that the superior inference runtime performance of certain APIs is often a byproduct of optimizations within the API rather than the underlying model. Our code is open sourced at https://github.com/stanford-crfm/helm-efficiency.

## 1 Introduction

Large language models (LLMs; Devlin et al., 2019; Brown et al., 2020; Rae et al., 2021; Lieber et al., 2021; Smith et al., 2022; Black et al., 2022; Chowdhery et al., 2022; OpenAI, 2023) have grown in size by almost four orders of magnitude in recent years, achieving state-of-the-art accuracy on traditional tasks like question answering and summarization (Zellers et al., 2019; Hendrycks et al., 2020; Zhang et al., 2023). LLMs display many exciting new capabilities as well, like reasoning about the physical world (Bisk et al., 2020), solving math problems (Cobbe et al., 2021; Li et al., 2023), and generating code (Chen et al., 2021), to name a few. To capitalize on these capabilities, several organizations offer access to LLMs through black-box APIs (OpenAI; AI21; Cohere) and many companies are deploying LLM-powered products at scale like ChatGPT, Bing, `jasper.ai` and Github Copilot (Reuters; Microsoft; Scale VP).

When building models, both users and developers must balance the benefits of new capabilities against the costs of scale. Recent efforts have begun to systematically evaluate and compare the downstream

---

[*]Work done while author was at Microsoft Research.

37th Conference on Neural Information Processing Systems (NeurIPS 2023).

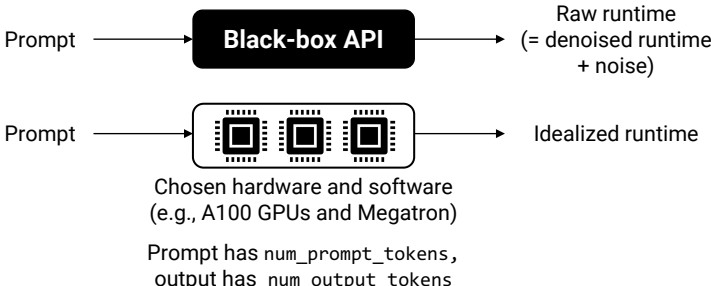

Figure 1: Comparison of raw runtime to the two runtime metrics proposed in this work.

task accuracies of LLMs (Brown et al., 2020; Rae et al., 2021; Liang et al., 2023; Srivastava et al., 2023), while others have examined the massive energy, financial, and computational costs of model *training* (Cao et al., 2020; Henderson et al., 2020; Strubell et al., 2019; Bender et al., 2021; Patterson et al., 2021; Bommasani et al., 2021, §5.3). However, few have considered the trade-offs of **inference efficiency vs. capability improvements** (Bommasani et al., 2023). This is important since model inference costs might far outweigh training costs for certain applications (e.g., ChatGPT had about 180 million unique visitors in August 2023 (Reuters, 2023)).

Our main contribution in this paper is a cost model for the end-to-end runtime of autoregressive generation using Transformer models with multi-head attention (Vaswani et al., 2017). Since autoregressive inference is composed of two stages that have vastly different computational properties (Cursor, 2023), the end-to-end runtime cannot be estimated directly from the total number of floating-point operations. We instead consider each of these two stages independently, and observe that inference runtime can be expressed as the sum of a parameterized piecewise linear function of the number of prompt tokens and a linear function of the number of output tokens under certain assumptions[2]. These parameters are specific to the target model and the software / hardware stack used. Our cost model can be efficiently fit to runtime profiles collected on either dedicated hardware or through black-box APIs, and allows us to estimate the runtime of a workload without running it in its entirety.

**Raw runtimes** of inference queries to black-box APIs are not inherently *comparable* across model providers since the API can include optimizations orthogonal to the model (e.g., caching, customized hardware, etc.) and be susceptible to performance variance (e.g., in our experiments, we found that heavy load can worsen raw runtime by up to $2\times$ for certain model providers). This makes it hard to gauge the inference efficiency of models on a level playing field, which can be important for model creators and researchers to understand the full *long-term costs* of various training decisions (e.g., model architecture / number of parameters). Raw runtime is still a good metric for end users who are directly impacted by slow (or fast) predictions.

We propose inference efficiency metrics that facilitate apples-to-apples comparisons across models. The main metric we propose is the **idealized runtime**, which is the runtime of an inference query if run on **a specified common software and hardware stack**. The idealized runtime can also be used to compute idealized energy and dollar costs to take into account the cost of the hardware used to serve the model. All of these idealized metrics can be estimated efficiently with our cost model.

Using these metrics, we conduct a novel analysis of inference efficiency-capability tradeoffs for various Transformer models available through black-box APIs (§5.3). Our analysis reveals several insights about the relative performance of these models. For example, the vanilla `OpenAI/davinci` model is often on the Pareto frontier of the efficiency-capability trade-off landscape when using raw runtime as the efficiency metric on 4 NLP scenarios covering sentiment analysis, question answering and classification. However, this efficiency appears to come from optimizations within the API rather than the model; `OpenAI/davinci` is consistently not on the idealized runtime Pareto frontier. Overall, we found the idealized metrics to be a useful tool for model creators and researchers to understand the true inference runtime, energy, and dollar costs that result from a particular model architecture and training process.

---

[2]Our cost model focuses on dense Transformer models for now. We believe this is a reasonable compromise since modern text generation APIs are powered almost exclusively by such models. The cost model needs to be slightly modified to cover popular variants like mixture-of-expert (MoE) models. We also mainly concentrate on small to medium context windows in this paper; see §3.1.2 for a full discussion.

## 2 Autoregressive Inference of Language Models

The input to a language model is a sequence of *tokens* (e.g., words). Autoregressive language models like GPT-3 (Brown et al., 2020) estimate the conditional probability $\Pr(x_i|x_{1:i-1})$ of a token $x_i$ given prefix tokens $x_1, x_2, \ldots, x_{i-1}$. During training, where we know all tokens in the training input a priori, the conditional probabilities $\Pr(x_1|\emptyset), \Pr(x_2|x_{1:1}), \Pr(x_3|x_{1:2}), \ldots, \Pr(x_s|x_{1:s-1})$ can be estimated in parallel, and thus only a single forward pass needs to be executed in every iteration before the backward pass. However, at inference time, outputs of the model need to be fed back in as inputs to generate subsequent outputs. In particular, a token $x_i$ is sampled from the conditional probability distribution obtained by running a forward pass through the model. Different sampling approaches can be used to obtain the token $x_i$ from the conditional probability distribution $\Pr(x_i|x_{1:i-1})$; common approaches include greedy sampling, random sampling with temperature annealing, nucleus sampling, and beam search. The process then needs to be repeated for the next token $x_{i+1}$ and so on. Consequently, inference through an autoregressive language model needs to perform *multiple* forward passes. This entire procedure differs from traditional inference for other models which require just a single forward pass.

Inference queries to language models are seeded with a prompt, which is a set of initial tokens $x_1, x_2, \ldots, x_p$ (we assume that the prompt has $p$ tokens). The conditional distribution $\Pr(x_{p+1}|x_{1:p})$ can then be computed through a forward pass. We call this the "prompt encoding" phase. Each subsequent generated token (sampled from $\Pr(x_{i+1}|x_{1:i})$ where $i > p$) needs its own forward pass through the model, which we term the "token generation" phase.

## 3 Cost Model for Autoregressive Inference Runtime

We now seek to derive a cost model $T(o, p; \theta)$ for the runtime of autoregressive inference of Transformer models (Vaswani et al., 2017). $p$ is the number of tokens in the input prompt, $o$ is the number of generated output tokens, and $\theta$ is a set of (learnt) parameters specific to a particular model, software, and hardware deployment.

In this section, we first specify the functional form of $T$ based on the number of floating-point operations in the prompt encoding and token generation phases. We do not use the total number of floating-point operations to directly estimate runtime since every floating-point operation does not have the same "cost": in particular, the prompt-encoding phase is compute-bound while the token-generating phase is memory-bandwidth-bound for small batch sizes[3] (Cursor, 2023; Yu et al., 2022). We then validate the derived expression of $T$ with experiments. Finally, we describe a procedure to estimate the $\theta$ parameters from profiled runtimes; the profiled runtimes can be from a dedicated server deployment, or from black-box APIs.

### 3.1 Derivation of Cost Model

To generate $o$ tokens, $o - 1$ additional forward passes are needed (the first token is generated in the prompt encoding phase). The runtime of generating $o$ tokens given a prompt with $p$ tokens is the sum prompt_encoding_time($p$) and output_generation_time($o$).

### 3.1.1 Number of Floating-Point Operations

To derive an expression for the end-to-end runtime of autoregressive inference, we first derive expressions for the number of floating-point operations required for each of the two steps.

In this paper, we are interested in the smallest possible inference runtime for a given query and assume inputs to the model are grouped into batches of size 1, but the batch size can be made larger in general to improve accelerator utilization and throughput at the cost of latency. Our cost model $T$ can be adapted to accommodate other batch sizes $b$.

---

[3]Depending on the number of output tokens generated and the prompt size, end-to-end autoregressive inference becomes largely memory-bandwidth-bound as well; the larger the number of generated output tokens, the lower the effective throughput since computation is in the memory-bandwidth-bound portion of the computation for longer.

**Prompt encoding.** Transformer models consist of many Transformer layers, which themselves are composed of an attention layer (which measures the "importance" of input tokens to each other) and a two-layer FFN in traditional formulations. As outlined in §A, the total number of operations that need to be run in the prompt encoding phase for a single prompt of size $p$ is $24ph^2l\left(1 + \frac{p}{6h}\right)$, where $l$ is the number of Transformer layers in the model and $h$ is the hidden size. $p$ is usually $\ll 6h$, so the number of compute operations needed to encode prompts is $24ph^2l$. We assume that the costs of projecting into vocabulary space in the output layer of the model and sampling the next token given the distribution $\Pr(x_{i+1}|x_{1:i})$ are cheap compared to the other operators in the Transformer layer (Narayanan et al., 2021).

**Output token generation.** When using language models autoregressively to generate new text, all operators in the transformer model must be performed incrementally. Concretely, the key, query, and value transformations in the self-attention layer need to be performed for just the new token, and self-attention scores need to be computed between the new token and all previous tokens. We can compute the number of floating-point operations needed per Transformer layer to perform these computations. Let $i$ be the number of tokens generated so far (i.e., we are trying to generate the $(i + 1)^{\text{th}}$ token, including the prompt). The total number of compute operations needed to generate the $(i + 1)^{\text{th}}$ token for a single prompt is $24h^2l + 4ihl = 24h^2l\left(1 + \frac{i}{6h}\right)$ (see §A.2 for details). If $i \ll 6h$, which is largely true in practice (e.g., for `OpenAI/davinci`, the maximum context length is 2048 and $h = 12288$), the number of floating-point operations to generate a new token is roughly independent of the number of tokens generated so far. When context windows become large (e.g., in excess of tens of thousands of tokens as seen in some newer models (Anthropic, 2023; MosaicML, 2023)), the cost of generating each new token is no longer independent of the number of tokens seen so far. In this paper, we focus on smaller context windows ($<$10k tokens) where $i \ll 6h$.

### 3.1.2 Final Parametric Form

Runtime can be expressed as the ratio of the number of floating-point operations and throughput:

$$\textsf{prompt\_encoding\_time}(p; \alpha) = \frac{(24h^2l)p}{\textsf{prompt\_encoding\_throughput}(p)} = \alpha_p p \tag{1}$$

$$\textsf{output\_generation\_time}(o; \beta) = \sum_{i=2}^{o} \frac{24h^2l}{\textsf{output\_generation\_throughput}(i)} = \beta(o-1) \tag{2}$$

Here, $\alpha_p$ is the average runtime per prompt token (different for different prompt sizes $p$) and $\beta$ is the runtime to generate each additional output token. $\textsf{prompt\_encoding\_throughput}$ initially increases as $p$ and arithmetic intensity (Williams et al., 2009) increase; we approximate $\textsf{prompt\_encoding\_time}(p)$ as piecewise linear. $\textsf{output\_generation\_throughput}$ is constant independent of the token being generated; this means $\textsf{output\_generation\_time}$ is a linear function of $o$. $\textsf{prompt\_encoding\_throughput}$ and $\textsf{output\_generation\_throughput}$ can differ by an order of magnitude or more.

We break up the space of all possible prompt sizes into disjoint intervals $(p_0, p_1], (p_1, p_2], \ldots, (p_{n-1}, p_n]$. $p_0$ is assumed to be the smallest possible prompt size of 0, and $p_n$ is assumed to be the largest possible prompt size (i.e., the maximum context window size supported by the model). For each interval $(p_j, p_{j+1}]$, we have a corresponding parameter $\alpha_j$ that needs to be estimated; $\alpha_j$ is the average runtime per prompt token when the prompt size is in the interval $(p_j, p_{j+1}]$.

Summing Equation 1 and Equation 2 gives us an expression for the end-to-end runtime[4]:

$$T(p, o; \theta = (\alpha, \beta)) = \textsf{prompt\_encoding\_time}(p, \alpha) + (o-1)\beta$$

$$= \sum_{j=0}^{n-1} \mathbb{1}(p_j < p \le p_{j+1})\alpha_j p + (o-1)\beta. \tag{3}$$

To support very large context windows, the above equation needs to be modified from the sum of a piecewise linear function of $p$ and linear function of $o$ to the sum of a piecewise quadratic function of $p$ and a quadratic function of $o$.

---

[4]For simplicity, we will use $\textsf{prompt\_encoding\_time}$ in all subsequent equations since piecewise linear functions are clunky to write.

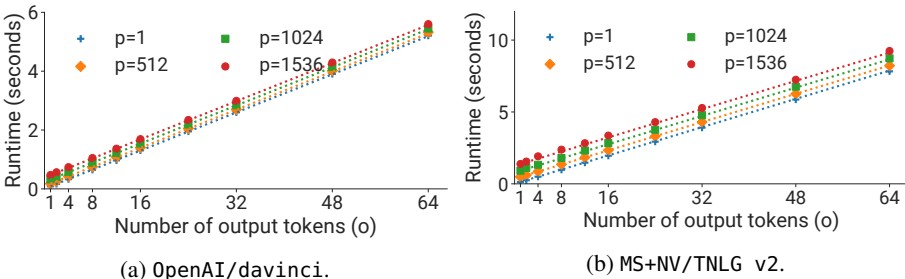

(a) `OpenAI/davinci`.  (b) `MS+NV/TNLG v2`.

Figure 2: End-to-end runtimes for different prompt sizes (shown in legend in terms of number of tokens) and models as the number of generated output tokens is varied using Megatron. We also show a dotted best-fit line.

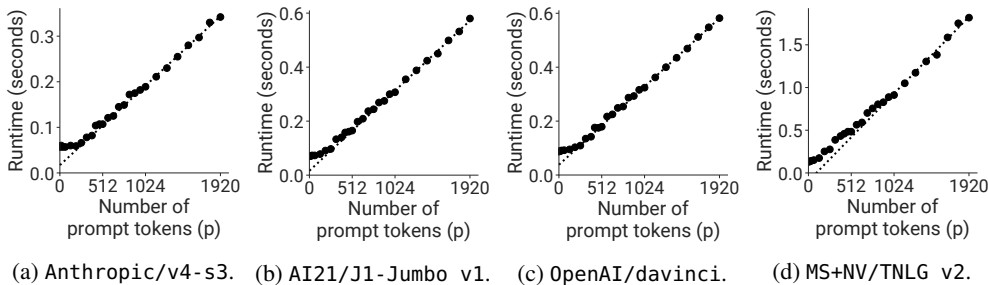

(a) `Anthropic/v4-s3`.  (b) `AI21/J1-Jumbo v1`.  (c) `OpenAI/davinci`.  (d) `MS+NV/TNLG v2`.

Figure 3: Prompt encoding runtimes versus prompt sizes. We also show a dotted best-fit line.

| Model (owner/name) | Provider | $h$ | $l$ | $n$ | # Params (B) | # GPUs×GPU type |
|---|---|---|---|---|---|---|
| `OpenAI/davinci` | OpenAI | 12288 | 96 | 96 | 175 | 8×80GB-A100 |
| `AI21/J1-Large v1` | AI21 Labs | 4096 | 32 | 32 | 6.7 | 1×80GB-A100 |
| `AI21/J1-Grande v1` | AI21 Labs | 5120 | 50 | 40 | 17 | 1×80GB-A100 |
| `AI21/J1-Jumbo v1` | AI21 Labs | 13824 | 76 | 96 | 178 | 8×80GB-A100 |
| `Cohere/XL v20220609` | Cohere | 8192 | 64 | 64 | 52 | 4×80GB-A100 |
| `Anthropic/v4-s3` | Anthropic | 8192 | 64 | 64 | 52 | 4×80GB-A100 |
| `MS+NV/TNLG v2` | Microsoft | 20480 | 105 | 128 | 530 | 24×80GB-A100 |
| `EleutherAI/GPT-J` | Together | 4096 | 28 | 16 | 6 | 1×80GB-A100 |
| `Yandex/YaLM` | Together | 10240 | 80 | 128 | 100 | 4×80GB-A100 |
| `BigScience/BLOOM` | Together | 14336 | 70 | 112 | 176 | 8×80GB-A100 |

Table 1: Models studied in this paper. We also specify the number of GPUs / GPU type used to estimate the default idealized runtimes. Different configurations are used with 32GB-V100 GPUs.

## 3.2 Empirical Validation

We can validate the above equations empirically on instantiations of state-of-the-art LLMs.

**Setup.** We use Megatron (NVIDIA), a high-performance GPU implementation of Transformer models with support for autoregressive inference[5]. For a given model, we used the minimum number of GPUs necessary to minimize cost. For example, `OpenAI/davinci` cannot fit on a single 80-GB A100 GPU; we use tensor model parallelism (Shoeybi et al., 2019) to ensure that the model parameters fit in GPU memory in such cases. Tensor model parallelism works well within a multi-GPU server (Narayanan et al., 2021) since expensive all-to-all communication collectives like all-reduce are limited to fast high-bandwidth NVLink. For even larger models like `MS+NV/TNLG v2`, we need other forms of parallelism like pipeline model parallelism in order to fit the model in GPU memory without poor scaling. We used NVIDIA HGX servers with 8 NVIDIA A100 SXM4 80GB GPUs; A100 GPUs were the fastest *widely available* GPU as of October 2022, when we did this work.

---

[5]CUDA version 11.5.0, Megatron commit hash `e156d2f` and `fp16` precision.

We evaluate 10 models, ranging in size from 6 to 530 billion parameters. Evaluated models are available in different ways: some were public via a commercial API (e.g., `OpenAI/davinci`, `AI21/J1-Jumbo v1`), some were private but the model owner provided research access for this effort (`Anthropic/v4-s3`, `MS+NV/TNLG v2`), and some were public and free (e.g., `BigScience/BLOOM`) and were run using the Together API[6]. We do not evaluate models with withheld model architecture details (e.g., `ChatGPT`). Table 1 shows the full set of evaluated models, along with the key hyperparameters released by the respective model owner that determine their size.

**Results.** Figure 2 shows the end-to-end runtime measured using the above setup, versus number of generated output tokens for different prompt sizes and models. We instantiate models based on reported architectures, but with random (untrained) parameters, as we only care about estimating runtime, and runtime is independent of the model's parameters given a prompt size and number of output tokens. We randomly sampled 4 prompt sizes ($\{1, 512, 1024, 1536\}$) from the space of all possible prompt sizes and 7 different number of output tokens ($\{1, 2, 4, 8, 16, 32, 64\}$). Runtime was averaged over 100 prompts of the same size. For each $p$, we compute a best-fit line using linear regression. We observe that the coefficients of determination ($R^2$) for the resulting time estimates are very close to 1.0 ($> 0.999$) for all models and conclude runtime shows a linear relationship with the number of output tokens for each prompt size (i.e., output_generation_time is a linear function of $o$).

Runtime also increases with prompt size. Figure 3 shows the prompt encoding time (i.e., runtime when the number of generated output tokens is set to 1) versus prompt size ($p$) for 4 models. We show prompt encoding runtimes for all $p \in \{1, 32, 64, 128, 192, \ldots, 1856, 1920\}$. Runtime and the prompt size have a roughly linear relationship, especially at large prompt sizes (shown by the dotted best-fit line). At small prompt sizes, $\alpha_p$ changes quickly; consequently, points deviate from the best-fit line on the left side of each figure.

### 3.3 Estimation Procedure

Equation 3 provides an expression for the end-to-end runtime of autoregressive Transformer LLMs for arbitrary prompt size $p$ and number of generated tokens $o$ that depends on $(m, s, h)$-specific parameters ($m$ is model, $s$ is software, $h$ is hardware). This suggests an efficient way of estimating $\theta = (\alpha, \beta)$ and consequently the runtime of a query with given prompt size and number of output tokens on a target system from a small number of profiled runtimes.

For each model and target system, we follow a two-step process. First, for each prompt size $p \in \{p_0, p_1, \ldots, p_n\}$, we profile the autoregressive Transformer LLM with numbers of output tokens equal to 1 to get the corresponding $\alpha_j = \text{runtime}_j / p_j$. Equipped with these $\alpha_j$ values, we leverage the fact that total_runtime$(p, o) - $prompt_encoding_time$(p)$ is a linear function in $o$ to fit a single linear regression model with $y = $ runtime difference and $x = o - 1$ to estimate slope $\beta$. In practice, we profile the full cross product of a set of prompt sizes and number of output tokens (i.e., $(p, o) \in \{p_0, p_1, \ldots, p_n\} \times \{o_1, o_2, \ldots, o_m\}$) and use the resulting runtimes to estimate $\alpha$ and $\beta$ for the target model and system.

## 4 Idealized and Denoised Metrics

We now propose two runtime metrics that can be efficiently estimated using the cost function $T$.

**Idealized runtime.** The idealized runtime is the runtime of an inference query assuming a particular model architecture, software and hardware implementation (e.g., NVIDIA A100 GPUs and Megatron respectively). It allows for the inference efficiency of models to be directly compared with each other.

$$T^{\text{idealized}}_{(m,s,h)}(p, o; (\alpha^{\text{idealized}}_{(m,s,h)}, \beta^{\text{idealized}}_{(m,s,h)})) = \text{prompt\_encoding\_time}(p, \alpha^{\text{idealized}}_{(m,s,h)})) + (o-1)\beta^{\text{idealized}}_{(m,s,h)}. \quad (4)$$

**Denoised runtime.** We also propose a runtime metric that factors out the noise from performance variation when using black-box APIs. We call this the denoised runtime; it is the best-case runtime of a particular query when given access to the same software and hardware used by the API provider.

$$T^{\text{denoised}}_{m \text{ on API } a}(p, o; (\alpha^{\text{denoised}}_{m \text{ on } a}, \beta^{\text{denoised}}_{m \text{ on } a})) = \text{prompt\_encoding\_time}(p, \alpha^{\text{denoised}}_{m \text{ on } a}) + (o-1)\beta^{\text{denoised}}_{m \text{ on } a}. \quad (5)$$

---

[6]`https://together.ai`.

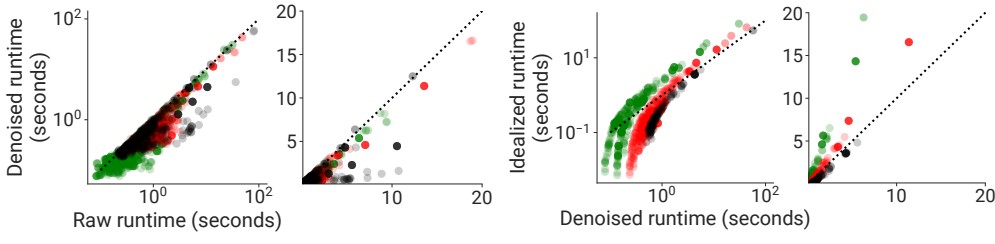

(a) Denoised runtime vs. raw runtime.  (b) Idealized runtime vs. denoised runtime.

Figure 4: Denoised vs. raw runtime and idealized vs. denoised runtime for various models across a range of queries along with a dotted $y = x$ line in log scale on the left and linear scale on the right (for both Figures 4a and 4b). OpenAI model points are shown in green, AI21 Labs points are shown in red, and points corresponding to other models are shown in black.

The runtime for generation *using a black-box API* is similar to Equation 3, but with an additional overhead term added to the prompt encoding and token generation runtimes. overhead captures the fixed costs of using an API to serve model predictions instead of using accelerators locally (e.g., round-trip latency of communicating with a remote API server) and performance variability (e.g., queuing delay or performance contention across requests). We quantify these overheads in §B.1.

Estimating runtimes of models run in a black-box system introduces additional complexity: runtimes using black-box APIs display higher variance relative to dedicated hardware (§B.1) due to performance contention, which can make it hard to estimate $\alpha$ and $\beta$. Since the variable performance overhead is a random variable $\eta \geq 0$, we run multiple trials for each $(p, o)$ pair in the profiling step and then perform linear regression using the *minimum* runtime (i.e., the runtime with minimum *variable* overhead across trials) for each $p$ and $o$. We see in §B.2 that this procedure is able to effectively factor out the performance variation observed in raw runtime measurements from black-box APIs.

**Idealized dollar cost.** Thus far, we have only focused on runtime. However, another important consideration is cost: larger models often require more accelerators just to fit the model in accelerator memory and so comparing runtimes alone might not be fair. We propose two metrics that explicitly take into account the number of accelerators for serving. Both metrics are derived from the idealized runtime. Unfortunately, we cannot similarly modify the denoised runtime since we do not know the number of chips used by the model provider. We compute the idealized dollar cost as $t_{(m,s,h)}^{\text{idealized}}$ (secs) $\times$ $n_{\text{accelerator } h} \times c_{\text{accelerator } h}$ ($/sec).

$n_{\text{accelerator } h}$ is the number of accelerators used at a time to serve a single request (1 if not using model parallelism, $> 1$ otherwise), and $c_{\text{accelerator } h}$ is the per-unit-time cost of the hardware $h$ (e.g., if $h$ is NVIDIA A100 GPUs, then $c_{\text{accelerator}}$ could then be the per-hour cost of renting an NVIDIA A100 GPU on AWS and the idealized dollar cost is the cost of serving the model on AWS A100 GPUs).

**Idealized energy cost.** Similar to work that has examined the energy cost of training (Henderson et al., 2020; Strubell et al., 2019; Patterson et al., 2021), we can estimate the idealized energy cost as $t_{(m,s,h)}^{\text{idealized}}$ (secs) $\times n_{\text{accelerator } h} \times p_{\text{accelerator } h}$ (W). $p_{\text{accelerator } h}$ is the power draw of hardware $h$. We can compare the idealized energy cost of running an inference query to the energy cost of training a model to better understand the number of inference queries needed to amortize training costs.

## 5   Results

In this section, we seek to answer the following: (a) Is the proposed methodology to estimate inference runtime of autoregressive Transformer models accurate and efficient compared to exhaustive profiling? (b) Can this method reveal interesting insights about models' efficiency-capability tradeoffs?

### 5.1   Accuracy of Runtime Estimation Procedure

We computed the coefficients of determination for runtimes of queries passed through black-box APIs for the models in Table 1. Despite performance variance, we see that the estimated runtimes using the methodology based on linear regression are fairly accurate ($R^2 > 0.9$), giving us further confidence that our cost model for runtime is accurate (full results in Table 2 in §B.2).

Figure 4a compares denoised runtimes to raw runtimes for a range of prompt sizes and number of generated output tokens. Experiments were conducted in September and October 2022. We observe that raw runtimes for the most part (96.6% of points) are greater than the estimated denoised runtimes (below the $y = x$ dotted line). Figure 4b is similar, but shows idealized runtime with A100 GPUs and NVIDIA's Megatron (Shoeybi et al., 2019) versus denoised runtime (same hardware and software setup as described in §3.2). In a number of cases, the idealized runtime is much lower than the denoised runtime, since the relevant API uses slower hardware and/or software implementations and other features like batching that sacrifice higher throughput for lower latency. For AI21 Labs models, idealized runtimes are greater than denoised runtimes 15.7% of the time. For OpenAI models, idealized runtimes are greater than denoised runtimes 64.2% of the time suggesting that they are using better hardware and / or a more optimized software stack. For all other model providers, idealized runtimes are always lower than the denoised runtimes, indicating that our hardware and software stack assumptions are fairly accurate.

## 5.2 Efficiently Evaluating Other Hardware or Software Optimizations

We can use the cost model proposed in this paper to efficiently evaluate the effectiveness of other hardware accelerators such as TPUs (Jouppi et al., 2017) or other generations of GPUs. As an example, Figure 11 in §B.4 shows how runtime and cost for various models is affected by using older NVIDIA V100 GPUs instead of NVIDIA A100 GPUs (the default hardware accelerator used in this paper). While we expect these GPUs to be slower, we can also reasonably expect them to be cheaper (due to lower per-hour costs (Amazon)). In practice, we find that this is *not the case*, suggesting V100 GPUs are slower and more expensive. This is partially because we often have to use double the GPUs to fit the model parameters in GPU memory, since V100 GPUs only have 32GB of device memory compared to 80GB on the A100 GPUs. This analysis requires profiling on the order of hours ($< 2$ hours for most models, depending on the number of $(p, o)$ values profiled) *once*, compared to hours *per benchmark* (depending on number of queries in the benchmark) for exhaustive profiling. As we show in §B.3, running a few calibration queries and then fitting the cost model to obtain $(m, s, h)$-specific parameters is much more efficient than exhaustive profiling. Our cost model also allows us to evaluate different software implementations and optimizations such as FasterTransformer (NVIDIA) or Flash-Decoding (Dao et al., 2023).

## 5.3 Efficiency-Capability Tradeoffs

We can now use the metrics proposed in this paper to evaluate the efficiency-capability tradeoffs of various language models accessible through black-box APIs. We focus on the few-shot evaluation setting adopted by BIG-Bench (Srivastava et al., 2023) and HELM (Liang et al., 2023). We consider four tasks in HELM: a sentiment analysis task (IMDB), two question answering tasks (MMLU [college chemistry] (Hendrycks et al., 2020) and BoolQ (Clark et al., 2019)), and a classification task (RAFT [terms of service] (Alex et al., 2021)). Figure 5 presents the results, with each row of graphs comparing average accuracy to a different efficiency metric. We highlight a few takeaways.

**Effect of model scale.** We observe that only a subset of the evaluated models fall on a Pareto frontier, with different models on the Pareto frontier for different tasks. This suggests that model scale alone does not predict model capabilities.

**Heterogeneous software / hardware.** The `OpenAI/davinci` model appears in the Pareto frontier for each benchmark when using raw runtimes but not the idealized metrics. This suggests that the OpenAI API implementation is more optimized than others: this could be due to a number of systems optimizations, such as query caching, or even model compression techniques like distillation, quantization or sparsification (Polino et al., 2018). Comparing these models on a level footing (same software and hardware) requires metrics that can factor out the effect of performance optimizations orthogonal to the model, such as idealized runtime.

**Model architecture design.** The relative positions of `BigScience/BLOOM` and `Yandex/YaLM` on the idealized cost and floating-point operations (+ model size) graphs are sometimes reversed: while `BigScience/BLOOM` achieves cheaper idealized cost (which takes into account the lower number of GPUs that `Yandex/YaLM` requires), `Yandex/YaLM` uses fewer floating-point operations. `BigScience/BLOOM`'s improved performance can be at least partially attributed to a more thorough

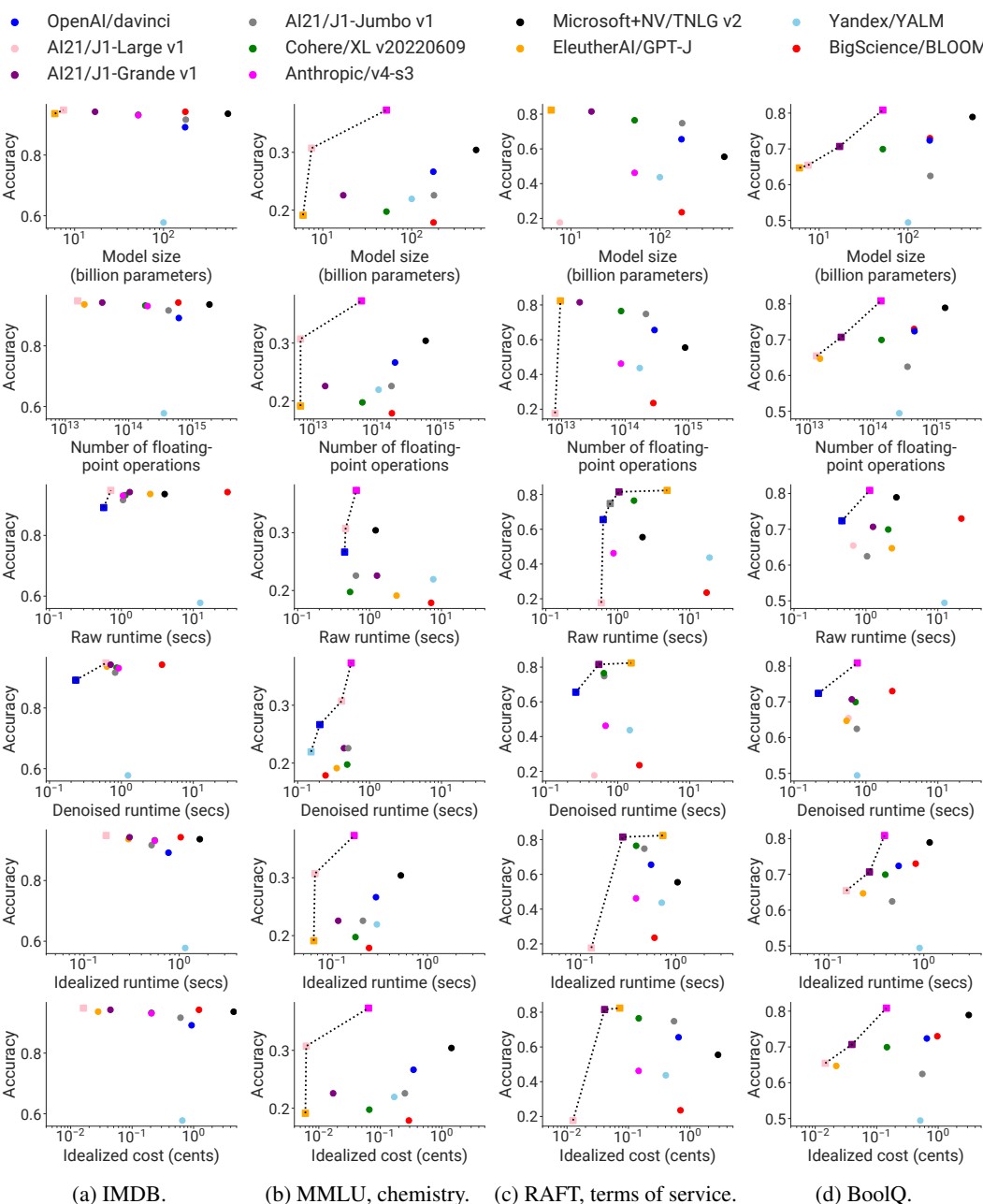

Figure 5: Capability vs. efficiency tradeoff graphs. Capability is shown as accuracy on the target task. Six efficiency metrics are shown: model size (billions of parameters), per-query number of floating-point operations (FLOPs), raw runtime, denoised runtime, idealized runtime (all in seconds), and idealized cost (in cents). Idealized metrics were estimated on the hardware and software setup described in §3.2. Metrics are averaged over all scenario instances. Models on the Pareto efficiency frontier are shown as squares with a black dotted line connecting the points.

search through model architectures for minimum runtime with a given number of floating-point operations in the forward pass (Scao et al., 2022).

**Run-to-run variance.** `AI21/J1-Grande v1` often achieves worse raw runtime than `AI21/J1-Jumbo v1` despite having $10\times$ fewer parameters, since the Grande model experiences higher performance variance (Figure 7). The idealized metrics make it easier to see the true efficiency-capability tradeoffs.

The denoised runtime metric can also help an end user reason through whether the observed inference performance is an artifact of performance contention.

**Cost comparison.** We can also compare these estimated *inference* costs to the costs charged by the black-box API provider. We observe that they are up to an order of magnitude lower than the charged actual costs. However, we note that these reported costs *do not* incorporate the significant cost of training models, which presumably gets amortized into the cost users pay with black-box APIs.

# 6 Related Work

A large body of work has looked at studying the impact of model scale on model capabilities.

**Scaling laws and other benchmarking efforts.** By fitting a curve to dozens of training runs, scaling laws (Kaplan et al., 2020) show how the size of a model in a model family affects the training and validation loss of these models. While these scaling laws are instructive, we also care about the capabilities of models along other axes beyond validation loss (e.g., are models robust; do they exhibit stereotypes?). Moreover, large language models have been shown to exhibit *emergent behavior* that cannot easily be expressed as a continuous function of scale (Wei et al., 2022). Similarly, even though model size is used as a proxy for training and inference runtime performance, it is not useful when trying to answer questions like "Can model $X$ meet a latency SLO of 100 milliseconds?". Consequently, empirical analysis is still needed to understand the capabilities of these models.

**Floating-point operations and other proxy metrics for efficiency.** The number of floating-point operations (FLOPs) required for the forward pass of a model has also often been used to estimate inference efficiency. While this is a fine approximation, it is not ideal for a couple different reasons. First, runtime does not correlate exactly with the number of FLOPs required (Scao et al., 2022). In particular, two operators with the same number of FLOPs could be executed with different efficiencies if one of the operators involves more memory accesses, preventing execution at peak device throughput. Second, as with model size, the number of FLOPs is hard to interpret. LLMs are often part of larger applications, and the performance requirements of these applications impose runtime constraints on LLM inference. It is hard to translate FLOPs to something actionable.

**Inference runtime estimation for other types of models.** Typically, inference for ML models is straightforward: an input of a particular size is passed through the model, in the process generating intermediate outputs and eventually a final prediction from a *single forward pass*. The sizes of intermediate outputs do not change from input to input, resulting in negligible runtime variance. This consequently makes inference runtime estimation easy. However, LLMs are different: while the hidden size does not change from input to input, the prompt size (in number of tokens) can be different for various inputs; runtime also depends on the number of output tokens generated.

**Carbon costs of ML computation.** Many papers (Canziani et al., 2016; Cao et al., 2020; Henderson et al., 2020; Strubell et al., 2019; Bender et al., 2021; Patterson et al., 2021) have discussed the importance of quantifying the cost of training models, both from an energy and emitted $CO_2$ perspective. This is often possible because model providers are open about details on training necessary to compute these metrics (Black et al., 2022; Patterson et al., 2021). While recent work has emphasized the need for considering inference-time efficiency (Henderson et al., 2020; Bommasani et al., 2021, §5.3), information on inference-time costs of LLMs is more scant for a multitude of reasons (e.g., optimizations powering an API might be part of a company's competitive advantage).

# 7 Conclusion

This work presents a systematic study of inference efficiency for autoregressive Transformer models accessible through black-box APIs. We showed both analytically and empirically that the inference runtime for these models is the sum of a piecewise linear function of the prompt size and a linear function of the number of output tokens, and designed an idealized runtime metric that can be estimated efficiently with minimal extra profiling. We are hopeful that our work helps model creators make better informed decisions about long-term model investments spanning training and serving.

## Acknowledgements

We thank the anonymous reviewers, Trevor Gale, Siddharth Karamcheti and Matei Zaharia for their help and feedback that improved this work. This work was supported in part by the AI2050 program at Schmidt Futures (Grant G-22-63429). This work builds on top of the Stanford HELM project; we are grateful to its many contributors. We also thank the following individuals at their respective organizations for the access, support, and/or credits required to evaluate their models:

- AI21 Labs: Opher Lieber, Barak Lenz, Dan Padnos, Yoav Shoham.
- Anthropic: Ben Mann, Jackson Kernion, Deep Ganguli, Jack Clark, Dario Amodei.
- Cohere: Lewis Stott, Ellie Evans, Bill McCartney, Aidan Gomez.
- Microsoft and the Turing Academic Program: Payal Bajaj, Barun Patra, Ahmed H. Awadallah, Saurabh Tiwary, Eric Horvitz.
- OpenAI: Lama Ahmad, Miles Brundage.

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

# Appendices

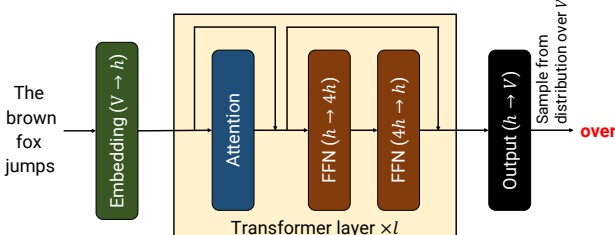

Figure 6: High-level schematic of a Transformer model with $l$ Transformer layers generating text at inference time given a prompt "The brown fox jumps".

# A   Derivation for Number of Floating-Point Operations

This paper focuses on language applications for Transformer models, where the model input is text. The input text is first preprocessed into a sequence of tokens (e.g., words) through a process called tokenization. Feature representations for each token (obtained by passing one-hot representations of the tokens through an embedding layer) are passed through multiple Transformer layers. Inputs to each Transformer layer are typically 3-dimensional tensors of shape $(b, s, h)$ where $b$ is the microbatch size (number of sequences), $s$ is the sequence length (number of tokens in each sequence), and $h$ is the hidden size (dimensionality of the model). For simplicity, we denote inputs as $X$.

Transformer layers in language models use self-attention to allow tokens to "interact" with each other. We assume multi-head attention; other forms of attention like multi-query attention (Shazeer, 2019) slightly change the analysis.

Self-attention is composed of the following operations:

- **Attention key ($K$), value ($V$), query ($Q$) transformations.** Given input $X$, we perform matrix multiplications $K = X \times W^K$, $V = X \times W^V$, and $Q = X \times W^Q$. $W^K$, $W^V$, and $W^Q$ are learned parameters.
- **Attention score computation.** Matrix multiplication $Q \times K^T$, followed by application of the softmax function to obtain score tensor $Z$. Each element $Z_{ij}$ is an importance score between query token $i$ and key token $j$. This is the primary mechanism that allows interaction across tokens in a sequence.
- **Attention over value computation.** Matrix multiplication of scores $Z$ by values $V$.

The subsequent two-layer feed forward network (FFN) consists of two linear layers (implemented as matrix multiplications). For most models, this involves multiplying the output of the self-attention layer by a matrix with dimension $h \times 4h$ and then multiplying the resulting output (after other operators like layer norm) by a matrix with dimension $4h \times h$. Figure 6 shows how these operators are connected in a typical "decoder-only" Transformer model.

We can now derive an expression for the number of floating-point operations in a typical forward pass through the model, using a similar form of analysis as prior work (Narayanan et al., 2021).

## A.1   Training

We use the same notation as before: $b$ is the microbatch size (number of sequences) and $h$ is the hidden size of the model. In practice, the self-attention layer computation described in §A is performed with different parameter matrices $W_i^K$, $W_i^V$ and $W_i^Q$. This is called running the self-attention layer with multiple *attention heads*. We assume that the Transformer model has $n$ attention heads.

$s$ is the sequence length in terms of number of tokens. Inputs $X$ to the Transformer layer have shape $(b, s, h)$. The Transformer layer's computation during training can then be reduced to the following matrix multiplication operations.

- Attention key, value, query transformations: These can be expressed as a single matrix multiplication of size: $(bs, h) \times (h, 3h)$. Output is of size $(bs, 3h)$.

- Attention score computation: $bn$ batched matrix multiplications (BMMs), each of size $(s, h/n) \times (h/n, s)$. Output is of size $(bn, s, s)$.

- Attention over value computation: $bn$ batched matrix multiplications of size $(s, s) \times (s, h/n)$. Output is of size $(bn, s, h/n)$.

- Post-attention linear projection: a single matrix multiplication of size $(bs, h) \times (h, h)$ to coalesce outputs of $n$ attention heads to a single per-sequence vector of size $h$. Output is of total size $(bs, h)$.

- Matrix multiplications in the MLP layer of size $(bs, h) \times (h, 4h)$ and $(bs, 4h) \times (4h, h)$. Outputs are of size $(bs, 4h)$ and $(bs, h)$.

Using the fact that a $(m, n) \times (n, k)$ matrix multiplication needs $2mnk$ floating-point operations, the total number of compute operations is to complete the forward pass through a Transformer layer during training is $24bsh^2 \left(1 + \frac{s}{6h}\right)$. A Transformer model typically has $l$ Transformer layers, resulting in a total of $24bsh^2l \left(1 + \frac{s}{6h}\right)$ floating-point operations in a single forward pass through the model. For most LLMs, $s \ll 6h$, meaning our expression for the number of floating-point operations in a single training forward pass can be simplified to $24bsh^2l$.

## A.2 Autoregressive Inference

We can similarly compute the number of floating-point operations needed to generate a single output token during autoregressive inference. $i$ is the number of tokens generated so far (i.e., the $(i + 1)^{\text{th}}$ token, including the prompt, needs to be generated next). The operators to be run in each Transformer layer in this phase are:

- Attention key $(K)$, value $(V)$, query $(Q)$ transformations: These can be expressed as a single matrix multiplication of size $(b, h) \times (h, 3h)$.

- Attention score computation: $bn$ batched matrix multiplications (BMMs), each of size $(1, h/n) \times (h/n, i)$ (only $Q$ value for the latest token is used; $K$ and $V$ values accumulated over all tokens so far).

- Attention over value computation: $bn$ batched matrix multiplication of size $(1, i) \times (i, h/n)$.

- Post-attention linear projection: a single matrix multiplication of size $(b, h) \times (h, h)$.

- Matrix multiplications in the MLP layer of size $(b, h) \times (h, 4h)$ and $(b, 4h) \times (4h, h)$.

Consequently, the total number of compute operations needed to generate the $(i + 1)^{\text{th}}$ token is $24bh^2l + 4bihl = 24bh^2l \left(1 + \frac{i}{6h}\right)$.

# B  Additional Results

In this section, we show additional results. In §B.1, we quantify the performance variation of using black-box APIs; in §B.2, we provide additional evidence that the estimation procedure described in §3.3 is accurate; in §B.3, we quantitatively show the cost benefits of our procedure compared to exhaustive profiling; and in §B.4, we show a case study of the type of quick analysis that this procedure can enable.

## B.1  Performance Variation in Black-Box APIs

We first measure the performance variation seen while using black-box text generation APIs.

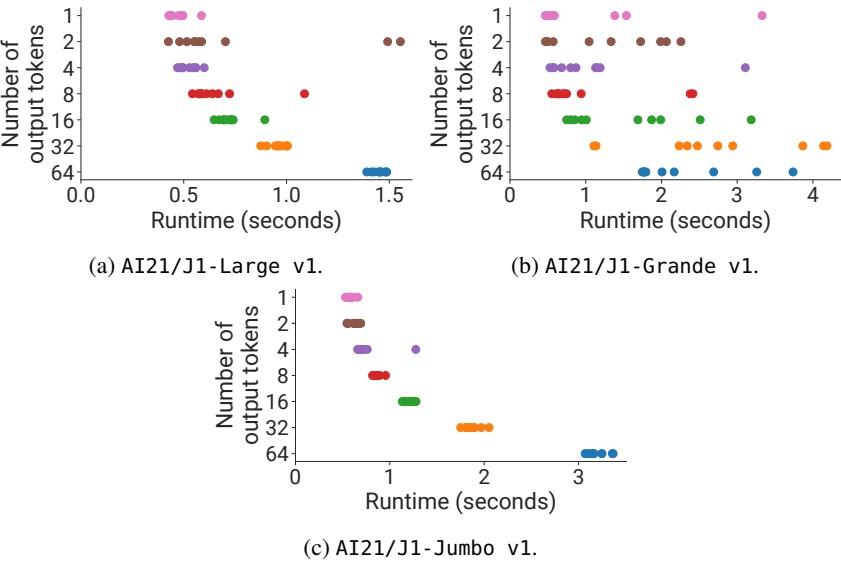

Figure 7: Per-instance runtimes using black-box APIs to access LLMs for multiple instances (prompt size, $p = 512$).

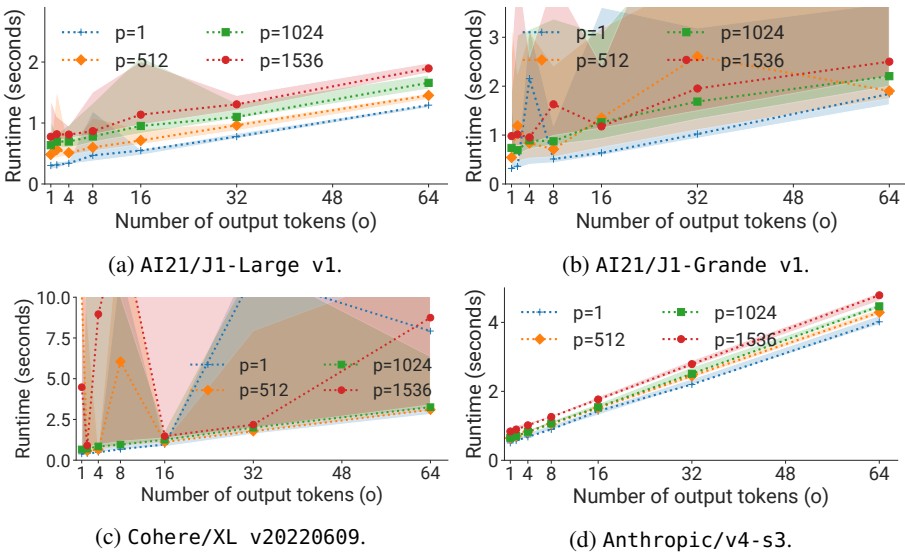

Figure 8: Median (dotted line) and min / max (lower and upper boundaries of shaded region) end-to-end runtimes across 5 trials for different prompt sizes (shown in legend in terms of number of tokens) as the number of generated output tokens is varied using black-box APIs. End-to-end runtimes show variation across the 5 trials for a given prompt size and number of output tokens.

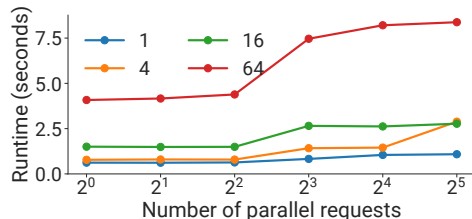

Figure 9: Minimum runtime across 10 trials as number of parallel queries increases for the `Anthropic/v4-s3` model. Prompt size is 512 tokens and the number of output tokens is varied (shown in legend). Experiment was run in 10/2022.

**Variation of runtimes across trials.** To better quantify performance variability when using black-box APIs, we run multiple trials of synthetic queries where we control the size of the prompt and the number of generated output tokens. Figure 7 and Figure 8 shows per-trial runtimes for different model offerings; Figure 7 shows runtimes for models from the same model provider (AI21). Unless otherwise noted, all experiments in this paper were run in September or October 2022 with the latest API versions available at the time.

We see discernible performance variance across multiple trials for different models, across prompt sizes and number of generated output tokens. Certain models experience higher performance variability: Figure 7 shows `AI21/J1-Grande v1` has much higher performance variance than `AI21/J1-Large v1` or `AI21/J1-Jumbo v1` (larger spread among points for a query of given size). `AI21/J1-Grande v1` has an average coefficient of variation of about 0.55 compared to much smaller coefficients of variation ($\sim$0.2) for the other AI21 models. Even for models with lower spreads (e.g., `AI21/J1-Large v1`), we see that outlier points can have as much as $3\times$ higher runtime.

Figure 8 also interestingly shows that this performance variance can obfuscate the linear relationship between model runtime and number of output tokens.

**Variation of runtimes with load.** To understand the impact of load on performance contention and end-to-end runtime, we measured query runtime as we increase the number of queries sent in parallel to the various black-box APIs. Figure 9 shows runtime versus number of parallel queries for different numbers of output tokens and a fixed prompt size of 512 tokens for the `Anthropic/v4-s3` model. We observe as much as a $2\times$ increase in runtime, indicating that load can lead to increased contention on API servers and consequently increased observed runtime.

## B.2 Accuracy of Runtime Estimation Procedure

| Model (owner/name) | $R^2$ |
|---|---|
| OpenAI/davinci | 0.976 |
| AI21/J1-Large v1 | 0.987 |
| AI21/J1-Grande v1 | 0.916 |
| AI21/J1-Jumbo v1 | 0.995 |
| Cohere/XL v20220609 | 0.995 |
| Anthropic/v4-s3 | 0.997 |

Table 2: Models and coefficient of determination ($R^2$) of time estimates for end-to-end text generation for various models using black-box APIs.

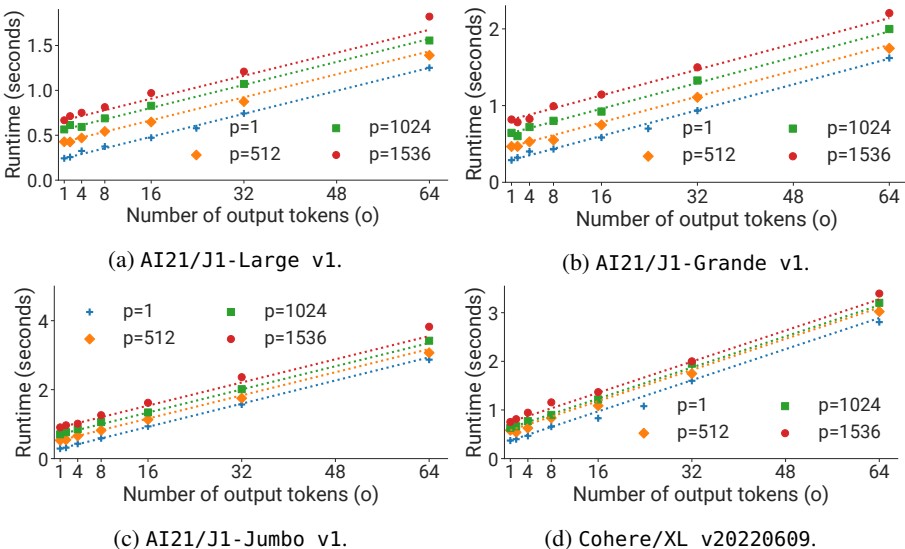

(a) AI21/J1-Large v1.

(b) AI21/J1-Grande v1.

(c) AI21/J1-Jumbo v1.

(d) Cohere/XL v20220609.

Figure 10: Minimum end-to-end runtimes for different prompt sizes (shown in legend in terms of number of tokens) as the number of generated output tokens is varied using black-box APIs, along with a best fit line estimated using the linear regression method described in §3.3 of this paper.

| Model (owner/name) | Metric | prompt_encoding_time ($p = 512/1024/1536$) | Per-output-token generation time ($\beta$) |
|---|---|---|---|
| OpenAI/davinci | $t^{\text{idealized}}_{(m,\text{ Megatron, A100})}$ | 0.178 / 0.323 / 0.476 | 0.081 |
| | $t^{\text{denoised}}_m$ | 0.045 / 0.033 / 0.142 | 0.030 |
| AI21/J1-Grande v1 | $t^{\text{idealized}}_{(m,\text{ Megatron, A100})}$ | 0.097 / 0.190 / 0.298 | 0.038 |
| | $t^{\text{denoised}}_m$ | 0.172 / 0.351 / 0.519 | 0.021 |
| AI21/J1-Jumbo v1 | $t^{\text{idealized}}_{(m,\text{ Megatron, A100})}$ | 0.164 / 0.310 / 0.465 | 0.064 |
| | $t^{\text{denoised}}_m$ | 0.268 / 0.463 / 0.655 | 0.042 |
| Anthropic/v4-s3 | $t^{\text{idealized}}_{(m,\text{ Megatron, A100})}$ | 0.108 / 0.189 / 0.279 | 0.054 |
| | $t^{\text{denoised}}_m$ | 0.193 / 0.191 / 0.380 | 0.057 |

Table 3: Models and estimated prompt encoding times / per-output-token generation times (in seconds) for $t^{\text{idealized}}_{(m,\text{ Megatron, A100})}$ and $t^{\text{denoised}}_m$.

We computed the coefficients of determination for runtimes of queries passed through black-box APIs; Table 2 and Figure 10 show the results. Despite performance variance, we see that the estimated

runtimes using the methodology based on linear regression are fairly accurate ($R^2 > 0.9$), giving us further confidence that our cost model is accurate.

Table 3 compares the learnt performance parameters for $t^{\text{idealized}}_{(m, \text{ Megatron, A100})}$ and $t^{\text{denoised}}_m$ for a subset of the considered models. As noted above, the estimated "(Megatron, A100) idealized" parameters for the AI21 Labs and OpenAI models are higher than the estimated denoised parameters, indicating that both these providers use optimizations not present in the software stack we considered.

## B.3    Comparison to Exhaustive Profiling

We can model the exhaustive profiling costs for computing idealized runtime and denoised runtime separately. For the idealized runtime, we can compare the cost of exhaustively running all queries in the local environment with estimating and then using the cost model (the approach proposed in §3.3). We run each query multiple times to get a reliable runtime measurement. In both cases, we need to run all queries through the API once in order to get the prompt size and number of generated tokens for each query.

We use the following notation:

- $t$: Number of trials.
- $Q$: Set of user queries.
- $\hat{Q}$: Set of calibration queries ($|\hat{Q}| \ll |Q|$).
- $c_{\text{API}}(q)$: Cost of invoking API on query $q$.
- $c_{\text{local}}(q)$: Cost of invoking local model on query $q$.

This produces the following expression:

$$\text{Idealized runtime savings} = \frac{t \cdot \sum_{q \in Q} c_{\text{local}}(q) + \sum_{q \in Q} c_{\text{API}}(q)}{t \cdot \sum_{q' \in \hat{Q}} c_{\text{local}}(q') + \sum_{q \in Q} c_{\text{API}}(q)}.$$

We can compute the savings concretely on the 4 HELM tasks evaluated in §5.3. We use the published per-token costs from OpenAI for `OpenAI/davinci` and the per-hour cost to rent an 8-A100 server from AWS (Amazon). With 50 trials per query (in practice, fewer trials are probably sufficient), we observe cost savings of 57× when using our cost model compared to exhaustive execution.

We can derive a similar expression to estimate the savings from using our approach for the denoised runtime, with the main difference being calibration queries now need to run through the black-box API, instead of on dedicated hardware.

$$\text{Denoised runtime savings} = \frac{t \cdot \sum_{q \in Q} c_{\text{API}}(q)}{t \cdot \sum_{q' \in \hat{Q}} c_{\text{API}}(q') + \sum_{q \in Q} c_{\text{API}}(q)}.$$

Using the same parameters as above, we get a savings of nearly 50× using the cost model compared to exhaustive execution (close to $t$ since $|\hat{Q}| \ll |Q|$).

## B.4   Case Study: Using the Cost Model to Compare Different Hardware

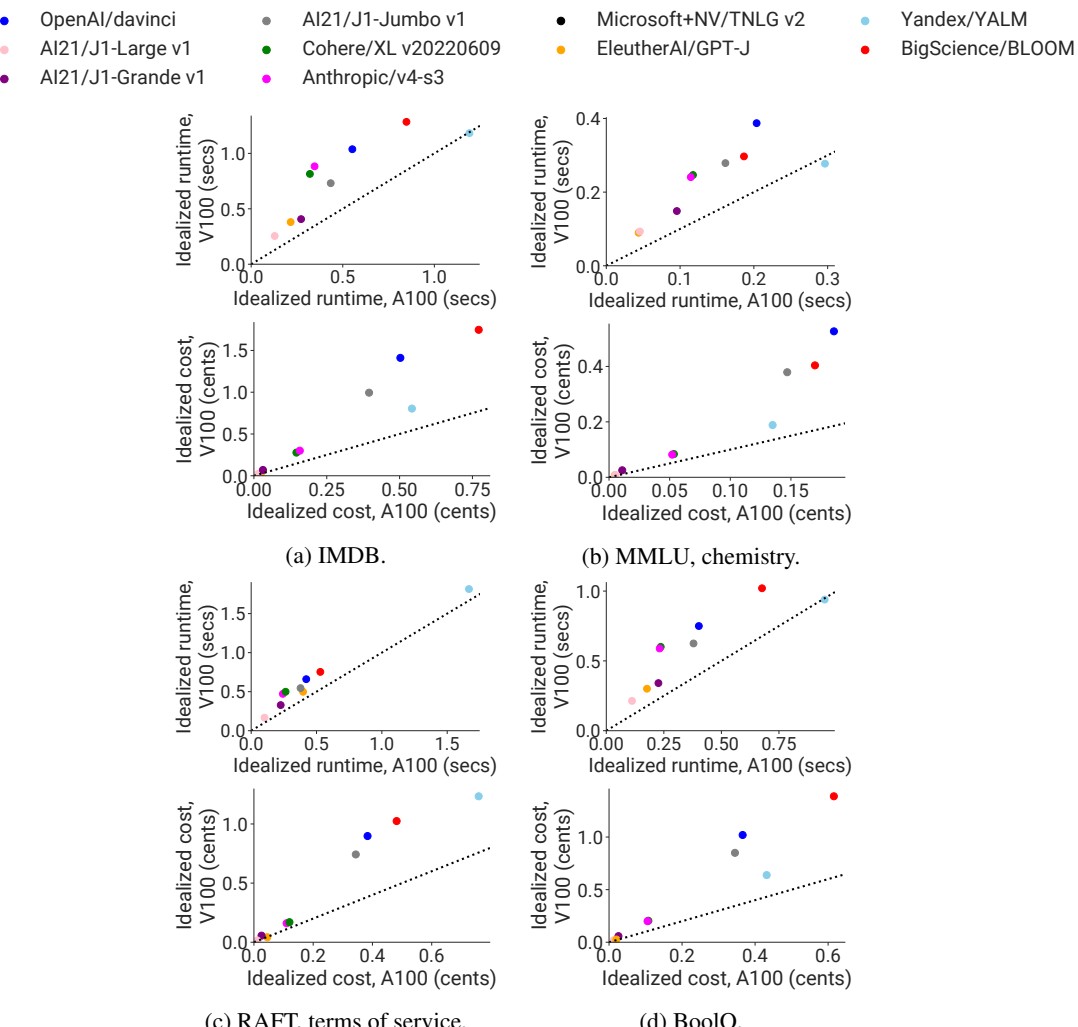

Figure 11: Comparison of idealized metrics estimated on different hardware.

Figure 11 shows a comparison between Megatron on NVIDIA A100 GPUs (the default configuration in this paper) to Megatron on NVIDIA V100 GPUs (an older generation of GPU). While we expect these GPUs to be slower, we can also reasonably expect them to be cheaper (due to lower per-hour costs (Amazon)). In practice, we find that this is *not the case*, suggesting V100 GPUs are slower and more expensive. This is partially because we often have to use double the GPUs to fit the model parameters in GPU memory, since V100 GPUs only have 32GB of device memory compared to 80GB on the A100 GPUs. This is an example of an analysis that we can quickly perform by just running calibration queries on V100 GPUs and fitting linear regression models to the resulting profiled runtimes.

