# OpenReview forum: "Cheaply Estimating Inference Efficiency Metrics for Autoregressive Transformer Models"
_NeurIPS.cc/2023/Conference — NeurIPS 2023 poster_

### Official Review · Reviewer_93KK · 2023-06-26

**Soundness:** 1 poor
**Presentation:** 2 fair
**Contribution:** 1 poor
**Rating:** 2
**Confidence:** 4

**Summary:**

The authours proposed a metric called `idealized runtime` to evaluate the inference efficiency among different LLMs as if they were run on the same standard hardware/software system. The main contributions can be summarized as following:

* the `idealized runtime` for a specific query with prompt length `p` and output length `o` on a target LLM can be collected by running the LLM on standard hardware/software (e.g., A100 and Megatron containers) **if target LLM's architecture is known/opensourced**
* the `idealized runtime` of transformer models can be modeled as a function linear to `o` and piecewise linear to `p` **if context window size is much smaller than embedding size of the model**. So with a few queries with different `p` and `o`, the coefficients of this function can be extracted by fitting the runtime of these queries to the function.
* with the fitted function, `idealized runtime` of queries can be estimated without running them exhaustively, so that the capacity of model (e.g., accuracy on a dataset/benchmark) can be compared against the pre-calculated/estimated `idealized runtime` of all the queries in the dataset to tradeoff/review scaling effect of LLMs.
* for closed models with api access, the authors proposed an alternative `denoised runtime` to approximate their idealized runtime.

**Strengths:**

* The authors pointed out the scaling law of LLMs should not just focus on accuracy vs FLOPs/Model size but accuracy vs inference latency/cost/power as well, which is a valuable point in LLM production.
* The authors proposed a simple way to estimate the idealized runtime of queries on standard hardware/software systems given prompt length and (projected) output length.

**Weaknesses:**

* The inference latency of a well known (open sourced) LLM architecture can be easily estimated by total ops of the model (e.g., those calculated in line 105/115) and divided by hardware effective (typically around 50% for LLMs on A100, https://arxiv.org/pdf/2104.04473.pdf or BLOOM paper) FLOP/s. The authors only mentioned about the drawback of this proxy briefly in line 300-308, without evidences based on target LLMs. Thus the significance of approximating these latencies via fitting instead of analytical calculation is quesitonable.

* The work makes too many assumptions that devalue its practice,
  1. context length << embedding size. This is true when the common context lenght is only 2K, but new development on LLMs have pushed them to 32K(GPT4)/64K (MPT)/100K(Claude), so this assumption and therefore the linear function approximation is no longer valid. With long context/prompt length, the analytical equation is probably more accurate.
  2. the `idealized runtime` is collected on standard Megatron container assuming model architectures are known. This limits the effectiveness of the work to open sourced models with efficient (distributed) implementations.
  3. the work assumes the pretrained/finetuned model is the final service/inference model. However in practice, the service model can be quantized/sparsified/distilled, so apple-to-apple inference latency comparision on pretrained models doesn't direclty translate to service model and the gap is typically huge, e.g., as shown by FIG3, the variance in idealized-runtime vs denoised-runtime is significant.

* The presentation of the work is not clear and sometimes repetitive/redudant.
  1. using FIG3 again as example, what's the diff of the two subplots in (a)/(b), it is not clarified and seems just different scales in axis. And as mentioned above, this FIG suggests for a couple models, the denoised-runtime is smaller than idealized-runtime, which is not a good evidence that idealized-runtime is a good approximation of roofline runtime.
  2. A lot of the equations are just rewrite/modifications to a simple linear function, and probably can be omitted or summarized in a table.

* Some of the conclusions are drawn without evidence, e.g., in line 257, the authors claimed "This suggests that scale alone does not predict model capabilities." Scaling law should be studied for the same model by varying model/dataset size. Comparing different model families with drastically different in pretraining dataset size/quality can not lead to any conclusion over model capacity over model size or inference latency.

* Overall the work is oversimplified by the assumptions above to a linear regression problem, which can be analytically calculated (and is commonly adopted, e.g., like Palm2 tech report) and becomes trivial to the audience of NeurIPS.

**Questions:**

Basically 1st point described in weakness. How far off is simple FLOPs estimated latency from the estimated `idealized runtime`, can you do a quantative study to support the necessity of doing this linear function fitting/approximation? And if you found the effective FLOP/s of an LLM is much lower/higher than 50%, can you study whether it is intrinsicaly due to the model (e..g, many non-dense ops) or the implementation.

**Limitations:**

The limitations are mainly the assumptions the authors made, as mentioned in weakness. The authors either didn't discuss them or without referring to SOTA LLM developments.

---

> ### Author Rebuttal · Authors · 2023-08-10
>
> Thank you for your review! We respond to your main comments and questions below.
>
> ---
>
> > The inference latency of a well known (open sourced) LLM architecture can be easily estimated by total ops of the model and divided by hardware effective (typically around 50% for LLMs on A100s) FLOP/s. The authors only mentioned about the drawback of this proxy briefly in line 300-308.
>
> This will not work well since the token generation part of the computation is not compute-bound, but is memory-bound; assuming 50% of peak compute throughput for this part of the workload will work _extremely poorly_ and leads to inaccurate runtime estimates (as pointed out below). Our point in lines 300-308 is slightly different: that the number of floating-point operations itself is not a good metric for practitioners since it is hard to interpret.
>
> ---
>
> > Context length << embedding size. New LLMs have pushed context lengths to 100k, so the linear function approximation is no longer valid.
>
> One thing to note here is that we want to compare the context length to ($6 \cdot \text{hidden\\_size}$). Models like GPT-3 have quite a large hidden size already (12288), so it is a reasonable assumption that context length is generally smaller than $6 \cdot 12288 = 73728$. Newer models like GPT-4 should have larger hidden sizes.
>
> For models where this approximation doesn’t quite hold, one can still fit a quadratic function to the runtime. This was not necessary when we initially did this work, but is not a dealbreaker going forward.
>
> We also note that some of these models (e.g., MPT with the larger context length) were released close to the NeurIPS deadline (~10 days before), and many current model offerings still have smaller maximum content lengths.
>
> ---
>
> > The idealized runtime is collected on standard Megatron container assuming model architectures are known. This limits the effectiveness of the work to open sourced models with efficient (distributed) implementations.
>
> This is not quite true. It is true that we need to know key model hyperparameters to do anything interesting here (e.g., the hidden size, number of layers, number of attention heads, etc.). But given these details, we can instantiate a model of the right size and use the same *open-source* software stack (that already has an efficient distributed implementation) to obtain runtime estimates. In fact, many of the models that we have already evaluated are not open-source (e.g., `OpenAI/davinci` and `AI21/J1-Jumbo`).
>
> ---
>
> > The work assumes the pretrained/finetuned model is the final service/inference model. However in practice, the service model can be quantized/sparsified/distilled.
>
> As noted in the paper: “For AI21 Labs models, idealized runtimes are greater than denoised runtimes 15.7% of the time. For OpenAI models, idealized runtimes are greater than denoised runtimes 64.2% of the time. For all other models, idealized runtimes are always lower than the denoised runtimes, [...]”. It is likely that OpenAI is implementing optimizations such as quantization, sparsification, distillation, additional parallelization, or better software optimization to extract inference runtime improvements; we want to emphasize that these optimizations are *orthogonal to our work*. We want to provide a metric that is apples-to-apples comparable across model providers; other model providers can also implement such optimizations to optimize their API offerings. The purpose of Figure 3 is to show that our estimated idealized runtimes are largely consistent with raw runtime measurements (and denoised runtime estimates).
>
> The denoised runtime metrics make more sense if one wants to compare the deployed models directly while taking out the effect of performance variation.
>
> ---
>
> > What's the diff of the two subplots in (a)/(b), it is not clarified and seems just different scales in axis.
>
> Apologies for not making this clear in the paper. The right subfigure in Figures 3a and 3b are a magnified version of the left subfigure in linear scale (instead of log-log scale). The y-intercept should be 0, and we will add a ytick to reflect this.
>
> ---
>
> > Some of the conclusions are drawn without evidence, e.g., in line 257, the authors claimed "This suggests that scale alone does not predict model capabilities."
>
> Apologies for the confusion. We were only referring to model scale (or the number of parameters in the model) here; it is of course true that the training dataset size affects model quality / capabilities, but nothing in our claim was meant to support or contradict that statement. We can change the phrasing in the paper to “This suggests that the number of model parameters alone does not predict model capabilities”. We can also add citations to indicate that other papers have also made this observation.
>
> ---
>
> > Overall the work is oversimplified; the analytical FLOPs for inference is commonly adopted, e.g., like Palm2 tech report.
>
> We respectfully disagree that this is “trivial”. The PaLM-2 report does have a discussion for optimal *training FLOPs* (6ND where D is the number of training tokens and N is the number of parameters in the model), not inference FLOPs.
>
> ---
>
> > How far off is simple FLOPs estimated latency from the estimated idealized runtime?
>
> The FLOPs-estimated latency is inaccurate.
>
> The effective FLOP/s is much lower than 50% because of the intrinsic structure of autoregressive inference: while training is largely compute-bound, autoregressive inference’s token generation stage is memory-bandwidth-bound. Depending on the number of output tokens generated and the prompt size, end-to-end autoregressive inference becomes largely memory-bandwidth-bound as well; the larger the number of generated output tokens, the lower the effective throughput since computation is in the memory-bandwidth-bound portion of the computation for longer. In our experiments, we found that effective inference FLOP/s for the various models studied in Section 5 varies from 15% to 41%.

---

> > ### Comment · Reviewer_93KK · 2023-08-16
> >
> > Thanks for your detailed explanations, though I am still not convinced on a few points,
> > > This will not work well since the token generation part of the computation is not compute-bound, but is memory-bound
> >
> > This is true if it is not batched inference (aka batch_size = 1). If you are only considering batch-size 1 inferencing, then equation (2) is misleading as it is approximating latency with total Ops ${\theta_{og}}$ divided by effective FLOPs (not sure what throughout means in the equation, I assume it refers to effective FLOPs). Instead it should depend on total memory traffic: parameters + activations ($12h^2l + O(iln)$ with n counting for num of heads in MHA. And the latency would be `total traffic / memory bandwidth` (using A100, it is roughly 60-70% of peak HBM2e  2TB/s), using BLOOM 176B (and assume i < h) as example, the roofline latency is 176G * 2Bytes / 1.3TB/s * 8GPU -> 34ms, not far from optimized real run (44ms: https://huggingface.co/blog/bloom-inference-pytorch-scripts).
> >
> > You can even approximate both prompts and generative inference latency (even with differnt batch-size) through max(total-Ops/Flops, Traffic/band-width) rather than limited to a linear model + interpolation).
> >
> > > so it is a reasonable assumption that context length is generally smaller than ... many current model offerings still have smaller maximum content lengths.
> >
> > Current offerings do not reflect the upcoming future (especially when the future is already happening) so it is not safe to assume "context length is generally **MUCH** smaller than 6H", maybe we can categorize this as limitations rather than weakness.
> >
> > > It is true that we need to know key model hyperparameters ... But given these details...
> >
> > Unfortunately many of the closed models don't even talk about `hidden size, number of layers, number of attention heads` at all, not mentioning there are architecture variants of MHA/MQA/GQA etc.
> >
> > > The purpose of Figure 3 is to show that our estimated idealized runtimes are largely consistent with raw runtime measurements (and denoised runtime estimates).
> >
> > I am not sure how this is considered largelyy consistent if you look at the right plot of Figure 3, the points don't lie close to y = x line.
> >
> > Overall I appreciate the concept of the metric `Idealized Runtime`, however I don't find its significance in practice. For well-known open-sourced model, the idealized inference latency can be closed form approximated by knowing the hardware system specs (https://www.cursor.so/blog/llama-inference), for closed models served in deployment environment, profiling representative samples with interpolation in between is a common practice, the procedure described in 3.4 is not innovative or insightful to a ML engineer with basic analytical skills. And the takeaway from section 5 is not clear, different models indeed have different inference latencies and differnt capacities, so while it can serve as a ranking board for application developers, it doesn't provide useful insights in terms of LLM training (like scaling laws) or service (like accuracy vs inference latency varied due to quatization/sparsification etc).

---

> > > ### Author Response · Authors · 2023-08-19
> > >
> > > Thank you so much for your comments. We respond inline (continued in next comment).
> > >
> > > ---
> > >
> > > > Batch_size > 1 is compute-bound.
> > >
> > > Batch_size > 1 will increase the arithmetic intensity but will still not make the end-to-end computation compute-bound (unless batch_size is very large, as in the offline setting, which we do not consider).
> > >
> > > ---
> > >
> > > > Analysis is misleading in batch_size=1 case.
> > >
> > > We used the same methodology as “Efficient Large-Scale Language Model Training on GPU Clusters” (Narayanan et al., SC 2021) where we reason about the total number of floating-point operations in the computation and then divide by the effective throughput of these operations (this is what $throughput_{og}(i)$ is in equation (2); since this part of the computation is memory-bandwidth-bound, $throughput_{og}(i)$ will be near the memory bandwidth number of the device). The purpose of this expression is only to provide intuition as to why the total runtime cost is linear in the number of generated output tokens (under certain assumptions); we can do a similar analysis with the number of memory operations and then divide by memory bandwidth of the device, as you mention in your comment.
> > >
> > > ---
> > >
> > > > The latency would be total traffic / memory bandwidth (using A100, it is roughly 60-70% of peak HBM2e 2TB/s), using BLOOM 176B (and assume i < h) as example, the roofline latency is 176G * 2Bytes / 1.3TB/s * 8GPU -> 34ms, not far from optimized real run (44ms: https://huggingface.co/blog/bloom-inference-pytorch-scripts).
> > >
> > > We agree that we can try to estimate runtime using device specs, but the analysis above a) makes assumptions (e.g., achieved memory bandwidth is 60-70% of peak; why not 80% or 50%), and b) even with these assumptions, the estimated runtime has about 25% error (34ms versus 44ms). Our method, on the other hand, doesn’t require these assumptions and also has much lower errors (R^2 scores of 0.999 or higher on A100s in the idealized runtime setting where we have direct access to the hardware and software) given a few calibration queries.
> > >
> > > ---
> > >
> > > > Current offerings do not reflect the upcoming future (especially when the future is already happening) so it is not safe to assume "context length is generally MUCH smaller than 6H".
> > >
> > > We agree that large context lengths are becoming more common. Again, as mentioned in our initial response, the NeurIPS submission deadline was <10 days after the launch of the MPT and Anthropic models with >65k maximum context lengths. To us, it does not seem like a realistic expectation to change course in such a short time frame. We did mention in our paper (lines 118-120) that context lengths are becoming larger.
> > >
> > > Having said that, we do believe we can extend our approach to handle long context lengths. The number of compute operations in the prompt encoding phase will start scaling quadratically with the prompt size, which means the runtime for prompt encoding will need to include a quadratic component. Similarly, the runtime for generating each additional token will increase slowly at large context lengths; consequently, instead of fitting a linear regression model between the runtimes and number of generated tokens, we will need to fit a quadratic regression model. We can include this analysis in the final version of this paper, and also validate the resulting expressions empirically.
> > >
> > > ---
> > >
> > > > Unfortunately many of the closed models don't even talk about hidden size, number of layers, number of attention heads at all, not mentioning there are architecture variants of MHA/MQA/GQA etc.
> > >
> > > We agree that it is unfortunate that these details are becoming more closed.
> > >
> > > ---
> > >
> > > > The points in Figure 3b don't lie close to $y = x$ line.
> > >
> > > $y=x$ will happen only if we know the exact hardware and software stack used by each model provider (impossible without insider information). On the other hand, we believe demonstrating that idealized runtime <= denoised runtime (when idealized runtime is estimated on relatively optimized hardware and software) shows that our estimated idealized runtimes are largely consistent with raw runtime measurements. In Figure 3b, idealized runtime <= denoised runtime for *all* black points (models from providers other than OpenAI and AI21 Labs), which is the majority of points shown. The fact that some  red and green points are above the $y=x$ line shows there is something *systematically incorrect* with our assumptions about the implementations used by OpenAI and AI21 Labs.

---

> > > > ### Author Response · Authors · 2023-08-19
> > > >
> > > > (Continuation of previous comment).
> > > >
> > > > ---
> > > >
> > > > > The procedure described in 3.4 is not innovative or insightful to a ML engineer with basic analytical skills.
> > > >
> > > > We respectfully disagree with this statement, especially the "basic analytical skills" part.
> > > >
> > > > ---
> > > >
> > > > > For well-known open-sourced model, the idealized inference latency can be closed form approximated by knowing the hardware system specs (https://www.cursor.so/blog/llama-inference), for closed models served in deployment environment, profiling representative samples with interpolation in between is a common practice.
> > > >
> > > > Thank you for linking to the blogpost (it seems that it was posted after the NeurIPS deadline). We agree that it is a useful resource, but want to point out that it does not perform interpolation of the runtimes of profiled calibration queries (as described in Section 3.4), but instead uses analytical expressions that rely on assumptions (e.g., “like FLOPs utilization, you can probably expect closer to 60-70% of that in inference workloads (1.3 TB/s)”) to obtain runtime estimates; even with these assumptions, these runtime estimates can be inaccurate (though the right order of magnitude). Our method, with a few calibration queries, can estimate runtimes more accurately.
> > > >
> > > > ---
> > > >
> > > > >  And the takeaway from section 5 is not clear, different models indeed have different inference latencies and different capacities, so while it can serve as a ranking board for application developers, it doesn't provide useful insights in terms of LLM training (like scaling laws) or service (like accuracy vs inference latency varied due to quantization/sparsification etc).
> > > >
> > > > As mentioned in our initial response: “We argue that the lack of clear winners is itself an interesting insight because it illustrates the complexity of evaluating the capabilities of LLMs: there is no one agreed upon task to evaluate the capabilities of LLMs, and no one LLM (at least among the ones we studied) dominates across a range of tasks. We want to emphasize that studying the capabilities of LLMs is not a direct goal of our work. Instead, we wish to augment such existing analyses by providing efficiency metrics comparable across different models, allowing us to provide additional context.” We believe there is follow-up work to be done to incorporate our idealized runtime metric into deeper analyses comparing different model offerings.
> > > >
> > > > Our methodology is general. We can estimate different versions of idealized runtimes (with various subsets of optimizations enabled; for example, a version of the idealized runtime with FlashAttention enabled and one without, or a version of the idealized runtime with sparsification enabled and one without) to characterize the impact of each optimization. We can also use the same methodology to efficiently estimate denoised runtimes (which capture all optimizations used by model providers in their API offerings) directly using API runtimes.

---

### Official Review · Reviewer_aZks · 2023-07-03

**Soundness:** 4 excellent
**Presentation:** 4 excellent
**Contribution:** 4 excellent
**Rating:** 8
**Confidence:** 4

**Summary:**

The authors proposed idealized runtime, a metric for measuring inference efficiency in LLMs, which measures the performance of models as if they were executed on a given hardware and software platform. Idealized runtime efficiently can be extended to estimate the idealized energy and dollar cost. The metric also takes into account the amount of hardware required to perform inference on a given model. Using these metrics, the authors compare ten state-of-the-art LLMs to demonstrate inference efficiency-capability tradeoffs. Overall, the authors aim to provide a more accurate and fair comparison of LLMs across different providers and to shed light on the tradeoffs between inference efficiency and capability.

**Strengths:**

1. The paper presentation is clear and the metric proposed is intuitive to understand.

2. The empirical study is thorough and examines the LLM inference efficiency from various perspectives.

3. The empirical findings demonstrate that the proposed metric can largely capture the efficiency-capability tradeoff in LLM inference.

**Weaknesses:**

1. The models on different tasks and metrics exhibit vastly different Pareto Frontiers (Figure 9). The cause of this variance hasn't been fully explored and explained in the work. Can these datasets/tasks properly evaluate the capabilities of the LLMs or is the variance due to something else?

**Questions:**

I have raised my concerns in the previous section.

**Limitations:**

The authors may further discuss the potential negative societal impact of their work.

---

> ### Author Rebuttal · Authors · 2023-08-09
>
> Thank you for your review! We respond to your comment on weaknesses below.
>
> ---
>
> > The models on different tasks and metrics exhibit vastly different Pareto Frontiers (Figure 9). The cause of this variance hasn't been fully explored and explained in the work. Can these datasets/tasks properly evaluate the capabilities of the LLMs or is the variance due to something else?
>
> We think that this is in part due to the difficulty in evaluating the capabilities of LLMs: there is no one agreed upon task to evaluate the capabilities of LLMs, and no one LLM (at least among the ones we studied) dominates across a range of tasks. Studying the capabilities of LLMs (or designing a single task that reasonably tests all of the facets of a LLM) is not a direct objective of our work. Instead, we wish to augment such existing analyses by providing efficiency metrics comparable across different models, allowing us to provide additional context.

---

### Official Review · Reviewer_a4zR · 2023-07-10

**Soundness:** 3 good
**Presentation:** 3 good
**Contribution:** 3 good
**Rating:** 6
**Confidence:** 5

**Summary:**

This paper aims to better understand the tradeoff between the inference efficiency and capability of LLM. It comes up with a metric called “idealized runtime” to fairly evaluate the inference efficiency of several popular LLMs, which is designed to factor out the irrelevant noises such as hardware, code stack, network latency and so on. With this metric, the authors analyised the efficiency and capability of LLMs and got some inpsiring insights.

**Strengths:**

First of all, I found the motivation of this paper very realistic and helpful to the community. We have been seeing lots of efforts made by the industry and academia to reduce the cost of the inference of LLMs, but how to compare those models from different parties fairly is yet to be figured out. This paper would be an inspiring exploration to this end. Besides, the analysis regarding the LLM inference is thorough and clear.

**Weaknesses:**

- Some technical details are kind of missing, which I will specify in the Question section.
- The concept of “idealized runtime” could be a bit unpractical because it requires one to implement and deploy the LLM on a very specific hardware and software setting, which may involve quite amount of proper engineering efforts.

**Questions:**

- As I understand, raw runtime and denoised runtime are derived from the API of model providers, and the idealized runtime is derived from the local cluster. Then how to interpret the impact of batch-size? As we know, the batchsize plays a quite import role in the padding, scheduling and I/O overhead during the inference, which could lead to very difference runtime. Since the batch size behind the API is unknown, how do we know if the differences between raw/denoised time and idealized time is actually simply dominated by the batch-size, instead of hardware and software?
- I wonder how to get the denoised runtime specifically? In the paper it says “we can run multiple trials in the profiling step and perform linear regression using the *minimum* obtained runtime across trials for each p and o”. So I guess it means you sent the same queries to a API n times, and chose the minimum runtime as denoised time? But how to do this sampling properly is important, for example, you may have quite different response time between morning and evening because of the traffic volume. And again, the impact of batch-size cannot be denoised here.
- In Figure3, why there are two figures in each subfigure? In the left figure of Fig3(b), why the y-intercept is not zero?
- Regarding the idealized runtime, it will be helpful if the authors list the detailed specifications of hardware and software setup, such as, CUDA version, Megatron version, tensor / pipeline parallel degrees, inter/intra node connections of GPUs, inference batch size.
- From Fig4, we can learn that the idealized time of different models actually aligns relatively well with the model size and FLOPS (except the little difference of BLOOM and YaLM), whereas the raw and denoised runtime is way more different than model size. How can we interpret this observation? Could we just use the model size and number of GPUs needed to roughly estimate the idealized time?
- Following the method described in section 3.4, we could estimate the linear regression function of prompting and generating time from an API, which is interesting. But in the experiment section, we only see the raw/denoised time in total, instead of breaking into prompting and generating time. I see a table in the Appendix about this data, but it would be nice to present a bit of them in the main paper as well. Otherwise the readers may wonder why do we need Section 3.4 if all the experiments are just end-2-end runtime.

**Limitations:**

The potential social impact is not discussed in the main paper.

---

> ### Author Rebuttal · Authors · 2023-08-09
>
> Thank you for your review! We respond to your main comments and questions below.
>
> ---
>
> > The concept of “idealized runtime” could be a bit unpractical because it requires one to implement and deploy the LLM on a very specific hardware and software setting, which may involve quite amount of proper engineering efforts.
>
> In this paper, we actually use an already-existing optimized open-source implementation (Megatron) on widely available hardware (a 8xA100 server with NVLink). To estimate a new model’s idealized runtime, one merely needs to instantiate a model of the right size (using appropriate command-line arguments) and run a few calibration queries to obtain runtimes on a few prompt sizes and number of output tokens. The resulting learnt regression parameters can then be used to estimate the idealized runtime for any arbitrary query with minimal engineering effort.
>
> ---
>
> > As I understand, raw runtime and denoised runtime are derived from the API of model providers, and the idealized runtime is derived from the local cluster. Then how to interpret the impact of batch-size? As we know, the batchsize plays a quite import role in the padding, scheduling and I/O overhead during the inference, which could lead to very difference runtime. Since the batch size behind the API is unknown, how do we know if the differences between raw/denoised time and idealized time is actually simply dominated by the batch-size, instead of hardware and software?
>
> We used a batch size of 1 since this gives us a lower-bound for latency for our assumed hardware and software. We agree that in practice, API providers might use a larger batch size to trade off latency for improved throughput. The denoised runtime is estimated using the minimum raw runtime seen across multiple runs of a given query; given enough runs, one of these runs should involve the query being executed as part of a batch of size 1.
>
> ---
>
> > I wonder how to get the denoised runtime specifically? In the paper it says “we can run multiple trials in the profiling step and perform linear regression using the minimum obtained runtime across trials for each p and o”. So I guess it means you sent the same queries to a API n times, and chose the minimum runtime as denoised time? But how to do this sampling properly is important, for example, you may have quite different response time between morning and evening because of the traffic volume. And again, the impact of batch-size cannot be denoised here.
>
> This is a good point. Your understanding is correct that naively, the denoised runtime is computed by running the same query multiple times and then choosing the minimum one. In practice, we only perform this procedure for a small set of calibration queries (Section 3.4), and then use the resulting profiled times to estimate the denoised runtime for all other queries. We make an implicit assumption that the smallest runtime across multiple runs of each calibration query corresponds to a batch of size 1. In general, we observed that the minimum raw runtime from our calibration queries have a largely linear relationship with the number of output tokens (see Figure 2 in the PDF attached to this rebuttal), similar to what we saw in our local cluster setting with Megatron.
>
> Runtime samples should be collected across different times of day in order to capture the effects of different traffic volumes..
>
> ---
>
> > In Figure 3, why there are two figures in each subfigure? In the left figure of Fig3(b), why the y-intercept is not zero?
>
> Apologies for not making this clear in the paper. The right subfigure in Figures 3a and 3b are a magnified version of the left subfigure in linear scale (instead of log-log scale). The y-intercept should be 0, and we will add a ytick to reflect this.
>
> ---
>
> > Regarding the idealized runtime, it will be helpful if the authors list the detailed specifications of hardware and software setup, such as, CUDA version, Megatron version, tensor / pipeline parallel degrees, inter/intra node connections of GPUs, inference batch size.
>
> Thank you, we will make this clear in the revised draft. We use CUDA version 11.5.0, Megatron version v2.6, tensor parallelism equal to the full number of GPUs used for inference (see Table 1 in the Appendix, attached as a part of the Supplementary Material) except for the MS+NV/TNLG v2 model, and batch size 1.
>
> ---
>
> > From Figure 4, we can learn that the idealized time of different models actually aligns relatively well with the model size and FLOPS (except the little difference of BLOOM and YaLM), whereas the raw and denoised runtime is way more different than model size. How can we interpret this observation? Could we just use the model size and number of GPUs needed to roughly estimate the idealized time?
>
> The model size and number of GPUs give a coarse directional estimate of the idealized runtime, but a) cannot answer questions such as “can queries be answered within a latency budget of 100ms?”, b) cannot directly predict the effect of hardware and software optimization (e.g., using H100 GPUs instead of A100 GPUs, or using FlashAttention instead of vanilla attention) on query runtime.
>
> ---
>
> > Following the method described in section 3.4, we could estimate the linear regression function of prompting and generating time from an API, which is interesting. But in the experiment section, we only see the raw/denoised time in total, instead of breaking into prompting and generating time. I see a table in the Appendix about this data, but it would be nice to present a bit of them in the main paper as well. Otherwise the readers may wonder why do we need Section 3.4 if all the experiments are just end-2-end runtime.
>
> This is a good point. Our main experiments focus on end-to-end runtime since we can only measure the raw end-to-end runtime using the black-box APIs. But we agree that it is interesting to present both of these separately, and we can add these to a revised version of the paper.

---

### Official Review · Reviewer_x2WX · 2023-07-23

**Soundness:** 3 good
**Presentation:** 3 good
**Contribution:** 3 good
**Rating:** 7
**Confidence:** 3

**Summary:**

"This paper introduces a range of metrics designed to compare the tradeoffs between inference efficiency and capability of autoaggressive Language Model (LLM) based on Transformers. Alongside the raw run-time metric, which can be influenced by infrastructure optimization and performance variance, the paper proposes additional metrics, namely idealized runtime, denoised runtime, idealized dollar cost, and idealized energy cost. These metrics enable a comprehensive comparison of various LLMs, even when they are served through different black-box APIs and hardware/software implementations.

**Strengths:**

* Originality: The paper introduces a novel analytical approach to estimate the inference efficiency of autoregressive Transformer models, overcoming challenges posed by models accessible only through black-box APIs and running on diverse hardware and software. This distinct proposal stands apart from previous analytical models designed for non LLMs. Moreover, it addresses the limitations associated with other proxies, such as model size and FLOPs.

* Quality and clarity: The proposal is presented with a rigorous analytical description and validated through empirical verification. The paper's structure is well-organized, and the content is conveyed in a clear and concise manner.

* Significance: The metrics, encompassing both the newly proposed and conventional ones, are effectively employed to compare the tradeoff between capability and efficiency. This comparison leads to several noteworthy insights, elaborated upon in Section 5.4.

**Weaknesses:**

There are a few minor formatting issues that require attention. For instance, in line 153, the notation of $o^2$ is confusing. The "2" is intended to refer to the footer, but it appears as if it denotes an exponent of 2.

**Questions:**

* In Figure 3, the denoised runtimes of OpenAI models are often lower than the idealized runtime. Could you provide further explanation for this phenomenon? Is it possibly related to the presence of better-optimized hardware and software infrastructure?

* Can you clarify which variables are being compared in Table 2? Are the comparisons made between Raw runtime and idealized runtime?

**Limitations:**

As mentioned in Section 3.1.1, the analytical model does not account for models with very large context windows (> 10,000); however, it is worth noting that these models are now becoming available.

---

> ### Author Rebuttal · Authors · 2023-08-09
>
> Thank you for your review! We respond to your main comments and questions below.
>
> ---
>
> > There are a few minor formatting issues that require attention. For instance, in line 153, the notation of o^2 is confusing. The "2" is intended to refer to the footer, but it appears as if it denotes an exponent of 2.
>
> Thank you for pointing this out, we will fix these formatting issues in the revised version of this paper.
>
> ---
>
> > In Figure 3, the denoised runtimes of OpenAI models are often lower than the idealized runtime. Could you provide further explanation for this phenomenon? Is it possibly related to the presence of better-optimized hardware and software infrastructure?
>
> Yes, the idealized runtime being larger than the denoised runtime suggests that OpenAI models use software and/or hardware implementations which are faster than the reference implementations we used to compute our runtime estimates (in particular, the Megatron library with the minimum number of A100 GPUs needed to fit the model / other tensors required for inference). We detail this behavior in the “Denoised Runtime” part of Section 4. Possible optimizations could include newer hardware (e.g., H100 GPUs), quantization, sparsification, distillation, or additional parallelization.
>
> ---
>
> > Can you clarify which variables are being compared in Table 2? Are the comparisons made between Raw runtime and idealized runtime?
>
> Yes, we compare raw runtime with idealized runtime in Table 2.
>
> ---
>
> > As mentioned in Section 3.1.1, the analytical model does not account for models with very large context windows (> 10,000); however, it is worth noting that these models are now becoming available.
>
> One thing to note here is that we want to compare the context length to ($6 \cdot \text{hidden\\_size}$). Models like GPT-3 have quite a large hidden size already (12288), so it is a reasonable assumption that context length is generally smaller than $6 \cdot 12288 = 73728$. One can only speculate that newer models like GPT-4 have larger hidden sizes.
>
> For models where this approximation doesn’t quite hold, one can still fit a quadratic function to the runtime. This was not necessary when we initially did this work, but is not a dealbreaker going forward.
>
> We also note that some of these models (e.g., MPT with the larger context length) were released close to the NeurIPS deadline (~10 days before), and a lot of the current model offerings still have much smaller maximum content lengths.

---

> > ### Comment · Reviewer_x2WX · 2023-08-16
> >
> > Thanks for the authors' response. I have no further questions and will keep my score.

---

### Official Review · Reviewer_rBmS · 2023-07-24

**Soundness:** 3 good
**Presentation:** 3 good
**Contribution:** 3 good
**Rating:** 4
**Confidence:** 4

**Summary:**

LLMs are dominating the NLP space at the moment, but efficient inference run-time and cost estimations have not been explicitly defined and tested. Raw run-times are not particularly useful across multiple services providing LLMs, as there is typically high variance overhead via the presence of black box prompt APIs. This paper proposes a few ways of estimating inference costs that attempt to provide an upper bound to the costs incurred by model computation. These metrics are demonstrated to be generally accurate and are cheap enough to be easily implemented, and the authors attempt to use them to generate new insights concerning various LLMs and generally large models across multiple services.

**Strengths:**

1. Very relevant, as LLMs are dominating conversations in the NLP space. Paper as a whole does a good job of making a case for the necessity of their work. Methodologies to estimate a separation of overhead costs from core model computation are inherently useful. Most, if not all, assumptions are fairly setup and seem correct.
2. The paper is very well-written. The problem setup is executed well and the initially proposed solution is very easy to follow. Explanations are mostly succinct, and most sections read as cogent.
3. The results look to be generally correct and explanations related to them are succinct. The results are made more relevant by the scale of the experiments, rendering the paper somewhat unique. The applied denoising factor estimation technique seems to roughly work in providing an upper bound for expected model computation time.


**Weaknesses:**

1. Some areas of the paper are a bit overexplained for this venue. Readers and reviewers of NeurIPS should, for example, be very familiar with autoregressive frameworks and how they generally work, or the forward pass design of generative AI in general.
2. Several early conclusions are mostly obvious. A lot of time is spent on specifying a procedure to estimate the number of floating point operations (and subsequently runtime based on throughput), but that isn't really all that interesting or new. For example, LLMs having roughly linear costs with respect to the size of their input (their quadratic attention costs are dwarfed) is probably well understood by your audience. Empirical proof related to this likely shouldn't be core to the paper.
3. Unclear if particularly new insights were generated from the results. There were largely no clear winners in cost-capability tradeoff analyses, there is no comparison to other evaluation techniques in the core body of the paper, and insights into when models fall or do not fall on the Pareto frontier of a given task do not seem initially useful.

**Questions:**

1. Some noticeable run-to-run variance was observed for certain models, but I did not spot exactly how it was dealt with beyond the selection of a minimum time. Could you elaborate on this point, especially for AI21/J1-Grande v1?
2. What would comparable exhaustive profiling cost a designer/user compared to your method? Can you provide some sort of comparison? Are there specific techniques you can compare your method to? Some materials from the appendix are mentioned, but this seems like it should be core content and included in an expanded Section 5.3.

---

> ### Author Rebuttal · Authors · 2023-08-09
>
> Thank you for your review! We respond to your main comments and questions below.
>
> ---
>
> > Some areas of the paper are a bit overexplained for this venue. Readers and reviewers of NeurIPS should, for example, be very familiar with autoregressive frameworks and how they generally work, or the forward pass design of generative AI in general.
>
> Thank you for this suggestion. We can defer lower-level details to the Appendix where appropriate.
>
> ---
>
> > Several early conclusions are mostly obvious. A lot of time is spent on specifying a procedure to estimate the number of floating point operations, but that isn't really all that interesting or new. For example, LLMs having roughly linear costs with respect to the size of their input (their quadratic attention costs are dwarfed) is well understood by your audience.
>
> Given that there hasn’t been a paper (or blogpost) describing these costs at length, we believe that our paper will be useful to the ML systems community, especially given the current interest in deploying efficient LLM inference services today. We have seen misconceptions on the internet (e.g., Twitter) regarding the relative cost of attention in the full model (we are withholding the link to preserve anonymity and double blind).
>
> ---
>
> > Largely no clear winners in cost-capability tradeoff analyses / no comparison to other evaluation techniques in the paper.
>
> We argue that the lack of clear winners is itself an interesting insight because it illustrates the complexity of evaluating the capabilities of LLMs: there is no one agreed upon task to evaluate the capabilities of LLMs, and no one LLM (at least among the ones we studied) dominates across a range of tasks. We want to emphasize that studying the capabilities of LLMs is not a direct goal of our work. Instead, we wish to augment such existing analyses by providing efficiency metrics comparable across different models, allowing us to provide additional context.
>
> We disagree that we do not compare to other efficiency evaluation techniques. We compare our metrics of idealized runtime and denoised runtime to raw runtime measurements as well as heuristics such as the model size and the number of floating-point operations, and show that these can be misleading (see Figure 4).
>
> ---
>
> > Some noticeable run-to-run variance was observed for certain models, but I did not spot exactly how it was dealt with beyond the selection of a minimum time. Could you elaborate on this point, especially for `AI21/J1-Grande v1`?
>
> As we see in Figure 8b, the `AI21/J1-Grande v1` model experiences significant variation across requests, which indicates that raw runtimes are an unreliable measure of API performance. To address this limitation of raw runtimes, we proposed the denoised runtime metric which estimates runtimes using the linear regression model detailed in Section 4. The denoised runtime estimates factor out performance variation by using the minimum runtimes observed across a small sample of trials as input to the linear regression model; the regression fitting method is also able to factor out noise in individual measurements.
>
> Figure 1 in the PDF attached to this rebuttal shows the min, median, and max end-to-end runtime for a subset of these models, and illustrates the degree of variation that we can see in raw runtimes. Figure 2 shows the minimum end-to-end runtime plotted against the number of output tokens for 5 models considered in our evaluation (including AI21/J1-Grande v1), along with the best-fit line estimated with our method. We can see that our procedure is able to factor out a lot of the performance variation that we observe in the raw runtime measurements.
>
> ---
>
> > What would comparable exhaustive profiling cost a designer/user compared to your method? Can you provide some sort of comparison?
>
> We can model the exhaustive profiling costs for computing idealized runtime and denoised runtime separately.
>
> For the idealized runtime, we can compare the cost of exhaustively running all queries in the local environment with running a comparatively small set of calibration queries as we propose in our paper (Section 3.4). We run each query multiple times to get a reliable runtime measurement. In both cases, we need to run all queries through the API once in order to get the number of prompt and output tokens for each query. We use the following notation:
>
> $x = \text{Number of trials}$.
>
> $Q = \text{Set of user queries}$.
>
> $\hat{Q} = \text{Set of calibration queries}\ (|\hat{Q}| \ll |Q|)$.
>
> $c_{\text{API}}(q) = \text{Cost to invoke API on query } q$.
>
> $c_{\text{local}}(q) = \text{Cost to invoke local model on query } q$.
>
> This gives the following expression:
> $$\text{Idealized runtime savings} = \frac{{x \cdot \sum_{q \in Q}{c_\text{local}(q)} + \sum_{q \in Q}{c_\text{API}(q)}}}{x \cdot \sum_{q' \in \hat{Q}}{c_{\text{local}}(q')} + \sum_{q \in Q}{c_\text{API}(q)}}.$$
>
> We can compute the savings concretely on the 4 HELM tasks evaluated in Section 5.4. We use publicly available per-token costs for the `OpenAI/davinci` model and the per-hour cost to rent an 8-A100 server from AWS. Assuming 50 trials per query (in practice, fewer trials are probably sufficient), this results in an overall savings of 57$\times$ when using our approach with calibration queries compared to exhaustive execution.
>
> We can derive a similar expression to estimate the savings from using our approach for the denoised runtime, with the main difference being calibration queries now need to be run through the black-box API, instead of locally.
>
> $$\text{Denoised runtime savings} = \frac{x \cdot \sum_{q \in Q}{c_\text{API}(q)}}{x \cdot \sum_{q' \in \hat{Q}}{c_{\text{API}}(q')} + \sum_{q \in Q}{c_\text{API}(q)}}.$$
>
> Using the same parameters as above, we get a savings of 48$\times$ using calibration queries compared to exhaustive execution (close to $x=50$, since both numerator and denominator in the above expression are dominated by the cost of running queries in $Q$).

---

> > ### Comment · Reviewer_rBmS · 2023-08-18
> >
> > Thanks for providing the responses. While the additional explanation is helpful, I remain slightly negative about the paper due to the limited new insights presented, and will keep my current score.

---

### Official Review · Reviewer_Jz6a · 2023-07-29

**Soundness:** 1 poor
**Presentation:** 2 fair
**Contribution:** 1 poor
**Rating:** 2
**Confidence:** 4

**Summary:**

This paper presents a new metric that attempts to more fairly evaluate machine learning model inference runtime performance cost, which the authors refer to as idealized runtime. The authors argue that the black-box nature of many large language models (LLMs) does not necessarily provide accurate estimates of inference cost using raw runtime results alone, due to several external factors (e.g., hardware used, compute resource contention, etc.). Thus, a goal of idealized runtime is to create a fair(er) inference evaluation metric for LLMs.

The authors analyze 10 state-of-the-art LLMs using their idealized runtime metric and discuss the efficiency tradeoffs between them.


**Strengths:**

- The paper is well-written
- The research topic of creating a somewhat universal metric to measure spatial and temporal overhead of LLMs on inference is relevant and timely
- The authors’ intuitions on focusing on compute operations (e.g., floating point instructions) and performing an inference runtime calculation that is not overtly computationally intense, is a good foundations for such research



**Weaknesses:**

While I agree with the authors that this topic is certainly worth deeply investigating, I have too many open concerns to recommend it for acceptance (at this time). Below I list two areas that I believe would need to be addressed before can be reasonably considered for publication.

- floating point operation overhead (a critical paper weakness): in Section 3.1.1 the authors discuss some of the concrete details used for their idealized runtime equation. Of note is the focus on deriving expressions “for the number of floating-point operations required for each of the two steps, and then use these to derive expressions for runtime.” While the number of floating point operations required to perform a single forward pass of inference is quite likely to an important datum for estimating inference runtime overhead, it is (in my opinion) just the first of many (perhaps dozens?) of factors that may be required to make an accurate estimation of inference overhead. There are several reasons for this, I list a couple below (the below list is not exhaustive).

(1) Not all floating point operations are spatially or temporally equivalent (e.g., a floating point ceiling or floor operation is notably less expensive than even a mathematical divide operation due to the execution “trap” that must exist for the divide, but not for the floor/ceil). Moreover, some floating point operations are orders of magnitude different in overheads (e.g., an FP divide is likely 10x-100x faster than a gather-scatter vector FP operation). To attempt to quantify all FP ops in a single computational class is logically unsound.

(2) The spatial and computational complexity of FP operations between different hardware compute classes varies significantly (often times more than 10x-100x). For example, GPUs can handle up to their channel size limit of FP data and continue to emit a fixed computation time for equivalent class FP operations. This is due to GPUs inherent SIMD properties. CPUs, on the other hand, emit approximately linearly increasing FP operation overhead as FP operation count increases. On the other hand, if the GPU channel size is exceeded even by a single datum, the GPU execution time may increase by 2x or more because it will generally require a second issuance of GPU block execution. Yet, CPUs only suffer roughly additive execution overhead when additional datum are added for FP operations. This means there is generally an inversely proportional runtime cost associated with FP operations and data between CPUs and GPUs. The paper seems to only focus on GPUs, which, in commercial practice, tends not to be representative (or at a minimum highly specialized) of which compute is used for inference.

- References: of the 50 or so references included by the authors, it appears that only around 10 of them are from peer-reviewed publications. The other 40 or so are from arxiv and websites. Moreover, the arxiv references are not necessarily recent (some date back over 5 years ago). The lack of peer-reviewed citation raises questions about the validity of substantiation as well as the authors’ understanding of the domain. A core problem with citing non-peer-reviewed published work is the lack of a fixed written artifact. Arxiv papers can have many versions as can website/webpages (which version are the authors referring to and how would a reviewer know this a priori?). I strongly encourage the authors to perform a more rigorous literature review and to substantiate their claims principally through peer-reviewed scientific research. It is a tall order to expect a reviewer to personally review / validate every non-peer-reviewed research paper to help verify its soundness and thus its appropriateness when used to substantiate the authors claims. Yet, not doing so would seem to imply that reviewers would need to implicitly “trust” the quality of such non-peer-reviewed artifacts. This seems antithetical to the evidence-based scientific research that is published at NeurIPS.


**Questions:**

None.

**Limitations:**

The authors seem to only focus on inference runtime overhead associated with GPUs (and a single class of GPU, as a reference point, Nvidia A100) and no other form of hardware compute. However, many other types of hardware are used to perform inference. Some of them are: CPUs, FPGAs, TPUs, CGRAs, etc. Moreover, each of these hardware classes have subclasses within that result in different temporal overhead and power footprint costs. None of these factors seem to be considered in the authors runtime equation.

---

> ### Author Rebuttal · Authors · 2023-08-09
>
> Thank you for your review! We respond to your main comments and questions below.
>
> ---
>
> > Floating point operation overhead (a critical paper weakness): in Section 3.1.1 the authors discuss some of the concrete details used for their idealized runtime equation. Of note is the focus on deriving expressions “for the number of floating-point operations required for each of the two steps, and then use these to derive expressions for runtime.” While the number of floating point operations required to perform a single forward pass of inference is quite likely to an important datum for estimating inference runtime overhead, it is (in my opinion) just the first of many (perhaps dozens?) of factors that may be required to make an accurate estimation of inference overhead.
>
> We want to clarify some misconceptions here.
>
> *Our method does not make the assumption that every floating-point operation has the same cost.* We certainly agree that not every floating-point operation has the same cost, and that the number of floating-point operations alone is insufficient to accurately estimate the runtime of a deep learning workload with many constituent operators. This is why we a) estimate prompt encoding (which is compute-bound) and token generation (which is memory-bandwidth-bound) runtime separately, b) use a piecewise linear function for prompt encoding runtime (since average throughput for matmuls increases with matmul size due to an increase in arithmetic intensity, as discussed in lines 145-150 in Section 3.2).
>
> Moreover, the metrics we propose (idealized runtime and denoised runtime) involve fitting a regression model to profiled *calibration runtimes directly* (Section 3.4) rather than relying on floating-point operation counts. The analytical expressions we show in Section 3.1 were intended to provide intuition and illustrate how the number of operations in each of these phases grow asymptotically, but our estimation method in practice does not directly use the number of floating-point operations. We can see from Figures 1 and 2 in the paper that the *measured end-to-end runtime for autoregressive inference* using existing optimized software and hardware implementations is linear in the number of generated output tokens and piecewise linear in the size of the prompt (only the best-fit line is estimated, the points are measured). We make similar observations when using raw runtimes of black-box APIs backed by similar Transformer models (Figure 2 in the PDF attached to this rebuttal).
>
> ---
>
> > The spatial and computational complexity of FP operations between different hardware compute classes varies significantly (often times more than 10x-100x). [...]  The paper seems to only focus on GPUs, which, in commercial practice, tends not to be representative (or at a minimum highly specialized) of which compute is used for inference.
>
> While other resource types (including CPUs, FPGAs, etc.) can be used for inference workloads, we believe GPUs are still widely used for inference at scale (e.g., see the “AI for everyone” section in https://news.microsoft.com/source/features/ai/how-microsofts-bet-on-azure-unlocked-an-ai-revolution/). However, we expect our method to work well for other hardware architectures as well given profiled runtimes on the appropriate hardware (the profiled runtimes on the target hardware should take into account the architectural differences outlined above); verifying this is an interesting area of future work.
>
> ---
>
> > References: of the 50 or so references included by the authors, it appears that only around 10 of them are from peer-reviewed publications. The other 40 or so are from arxiv and websites.
>
> Apologies for this. Some of the referenced arXiv papers have been published at conferences (e.g., FlashAttention appeared at NeurIPS 2022, BERT appeared at NAACL 2019), and we can update these references to point to the conference versions in an updated copy of the paper (Google Scholar often shows arXiv references as the default unfortunately). Many of the referenced papers that have not appeared at conferences are widely cited (e.g., the GPT-4 and PaLM technical reports, scaling laws paper, Switch-Transformer).
>
> ---
>
> > The authors seem to only focus on inference runtime overhead associated with GPUs (and a single class of GPU, as a reference point, Nvidia A100) and no other form of hardware compute. However, many other types of hardware are used to perform inference. Some of them are: CPUs, FPGAs, TPUs, CGRAs, etc. Moreover, each of these hardware classes have subclasses within that result in different temporal overhead and power footprint costs. None of these factors seem to be considered in the authors runtime equation.
>
> We used our procedure to estimate runtimes on Nvidia V100 GPUs as well, shown in Figure 10 in the Appendix (attached in the Supplementary Material). As mentioned above, verifying that our method works well on other architectures like TPUs is an interesting area of future work.

---

> ### Comment · Senior_Area_Chairs · 2023-08-18
> **@reviewer Jz6a: Please respond to authors' rebuttal**
>
> Given that there is a large divergence in the scores on this paper, it would be very helpful if you could read and respond to the authors' rebuttal. Does it address your concerns?

---

### Author Rebuttal · Authors · 2023-08-09

We thank all the reviewers for their time and effort in providing feedback on our paper, which will be useful in improving its quality.

All reviewers seem to agree that the problem addressed by the paper is important and well-motivated. However, the two lowest-score reviews seem to have serious misunderstandings of the paper:
- Our paper does not advocate using the number of floating-point operations to directly estimate end-to-end runtime (as reviewer Jz6a points out, the cost of each floating-point operation is not the same). Instead, we show the expressions for number of floating-point operations to provide intuition for our method (described in Section 3.4), which uses *runtimes of a set of calibration queries* to estimate runtime for any query (assuming a given hardware and software setup).
- Using the number of floating-point operations to estimate runtime works poorly. The effective FLOP/s for inference is much lower than 50% because of the intrinsic structure of the computation: while training is largely compute-bound, autoregressive inference’s token generation stage is memory-bandwidth-bound. Depending on the number of output tokens generated and the prompt size, end-to-end autoregressive inference becomes largely memory-bandwidth-bound as well; the larger the number of generated output tokens, the lower the effective throughput since computation is in the memory-bandwidth-bound portion of the computation for longer. In our experiments, we found that effective inference FLOP/s varies from 15% to 41%.

The main contribution of our paper is a general methodology to efficiently estimate the runtime of autoregressive inference of a dense Transformer model for a prompt of given size and number of output tokens on a given hardware and software stack, using profiling on a small set of calibration queries. The exact proposed method works best for prompts up to ~10k tokens for current-day models, but changing the output generation time from a linear model to a quadratic model could make runtime estimation accurate for even larger context sizes.

We respond to the other main concerns raised by reviewers in the per-reviewer comments.

---

### Decision · Program_Chairs · 2023-09-21

**Decision:**

Accept (poster)

**Comment:**

Paper proposes a method to efficiently estimate the runtime of autoregressive inference of a (dense) Transformer model for a prompt of given size and number of output tokens on a given hardware and software stack, using profiling on a small set of calibration queries.

All of the reviewers appreciate the concept of the metric Idealized Runtime, however some didn't find its significance in practice. Authors addressed several comments by the reviewers and majority found it to be convincing. Considering the lack of any previous attempt in this space, and the abundance of LLM services lately, studies like this one has the potential to redistribute the focus of the field onto inference cost/latency and predictability. Hence I found it to be significant to be shared via NeurIPS.